# EquiformerV2: Improved Equivariant Transformer for Scaling to Higher-Degree Representations

**Yi-Lun Liao**[1]    **Brandon Wood**[2]    **Abhishek Das**[2*]    **Tess Smidt**[1*]

[1]Massachusetts Institute of Technology    [2]FAIR, Meta    *Equal contribution

{ylliao,tsmidt}@mit.edu    {bmwood,abhshkdz}@meta.com
https://github.com/atomicarchitects/equiformer_v2

## Abstract

Equivariant Transformers such as Equiformer have demonstrated the efficacy of applying Transformers to the domain of 3D atomistic systems. However, they are limited to small degrees of equivariant representations due to their computational complexity. In this paper, we investigate whether these architectures can scale well to higher degrees. Starting from Equiformer, we first replace $SO(3)$ convolutions with eSCN convolutions to efficiently incorporate higher-degree tensors. Then, to better leverage the power of higher degrees, we propose three architectural improvements – attention re-normalization, separable $S^2$ activation and separable layer normalization. Putting this all together, we propose EquiformerV2, which outperforms previous state-of-the-art methods on large-scale OC20 dataset by up to $9\%$ on forces, $4\%$ on energies, offers better speed-accuracy trade-offs, and $2\times$ reduction in DFT calculations needed for computing adsorption energies. Additionally, EquiformerV2 trained on only OC22 dataset outperforms GemNet-OC trained on both OC20 and OC22 datasets, achieving much better data efficiency. Finally, we compare EquiformerV2 with Equiformer on QM9 and OC20 S2EF-2M datasets to better understand the performance gain brought by higher degrees.

## 1 Introduction

In recent years, machine learning (ML) models have shown promising results in accelerating and scaling high-accuracy but compute-intensive quantum mechanical calculations by effectively accounting for key features of atomic systems, such as the discrete nature of atoms, and Euclidean and permutation symmetries (Gilmer et al., 2017; Zhang et al., 2018; Jia et al., 2020; Gasteiger et al., 2020a; Batzner et al., 2022; Lu et al., 2021; Unke et al., 2021; Sriram et al., 2022; Rackers et al., 2023; Lan et al., 2022). By bringing down computational costs from hours or days to fractions of seconds, these methods enable new insights in many applications such as molecular simulations, material design and drug discovery. A promising class of ML models that have enabled this progress is equivariant graph neural networks (GNNs) (Thomas et al., 2018; Weiler et al., 2018; Kondor et al., 2018; Fuchs et al., 2020; Batzner et al., 2022; Brandstetter et al., 2022; Musaelian et al., 2022; Liao & Smidt, 2023; Passaro & Zitnick, 2023).

Equivariant GNNs treat 3D atomistic systems as graphs, and incorporate inductive biases such that their internal representations and predictions are equivariant to 3D translations, rotations and optionally inversions. Specifically, they build up equivariant features of each node as vector spaces of irreducible representations (or irreps) and have interactions or message passing between nodes based on equivariant operations such as tensor products. Recent works on equivariant Transformers, specifically Equiformer (Liao & Smidt, 2023), have shown the efficacy of applying Transformers (Vaswani et al., 2017), which have previously enjoyed widespread success in computer vision (Carion et al., 2020; Dosovitskiy et al., 2021; Touvron et al., 2020), language (Devlin et al., 2019; Brown et al., 2020), and graphs (Dwivedi & Bresson, 2020; Kreuzer et al., 2021; Ying et al., 2021; Shi et al., 2022), to this domain of 3D atomistic systems.

A bottleneck in scaling Equiformer as well as other equivariant GNNs is the computational complexity of tensor products, especially when we increase the maximum degree of irreps $L_{max}$. This limits these models to use small values of $L_{max}$ (e.g., $L_{max} \leqslant 3$), which consequently limits their performance. Higher degrees can better capture angular resolution and directional information, which is critical to accurate prediction of atomic energies and forces (Batzner et al., 2022; Zitnick et al., 2022; Passaro & Zitnick, 2023). To this end, eSCN (Passaro & Zitnick, 2023) recently proposes efficient convolutions

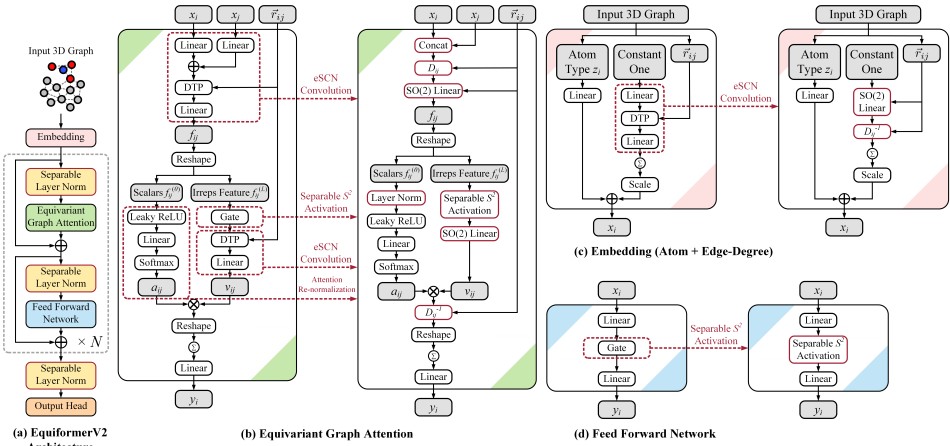

Figure 1: Overview of EquiformerV2. We highlight the differences from Equiformer (Liao & Smidt, 2023) in red. For (b), (c), and (d), the left figure is the original module in Equiformer, and the right figure is the revised module in EquiformerV2. Input 3D graphs are embedded with atom and edge-degree embeddings and processed with Transformer blocks, which consist of equivariant graph attention and feed forward networks. "$\otimes$" denotes multiplication, "$\oplus$" denotes addition, and $\sum$ within a circle denotes summation over all neighbors. "DTP" denotes depth-wise tensor products used in Equiformer. Gray cells indicate intermediate irreps features.

to reduce $SO(3)$ tensor products to $SO(2)$ linear operations, bringing down the computational cost from $\mathcal{O}(L_{max}^6)$ to $\mathcal{O}(L_{max}^3)$ and enabling scaling to larger values of $L_{max}$ (e.g., $L_{max}$ up to 8). However, except using efficient convolutions for higher $L_{max}$, eSCN still follows SEGNN-like message passing network (Brandstetter et al., 2022) design, and Equiformer has been shown to improve upon SEGNN. More importantly, this ability to use higher $L_{max}$ challenges whether the previous design of equivariant Transformers can scale well to higher-degree representations.

In this paper, we are interested in adapting eSCN convolutions for higher-degree representations to equivariant Transformers. We start with Equiformer (Liao & Smidt, 2023) and replace $SO(3)$ convolutions with eSCN convolutions. We find that naively incorporating eSCN convolutions does not result in better performance than the original eSCN model. Therefore, to better leverage the power of higher degrees, we propose three architectural improvements – attention re-normalization, separable $S^2$ activation and separable layer normalization. Putting these all together, we propose EquiformerV2, which is developed on large and diverse OC20 dataset (Chanussot* et al., 2021). Experiments on OC20 show that EquiformerV2 outperforms previous state-of-the-art methods with improvements of up to 9% on forces and 4% on energies, and offers better speed-accuracy trade-offs compared to existing invariant and equivariant GNNs. Additionally, when used in the AdsorbML algorithm (Lan et al., 2022) for performing adsorption energy calculations, EquiformerV2 achieves the highest success rate and 2× reduction in DFT calculations to achieve comparable adsorption energy accuracies as previous methods. Furthermore, EquiformerV2 trained on only OC22 (Tran* et al., 2022) dataset outperforms GemNet-OC (Gasteiger et al., 2022) trained on both OC20 and OC22 datasets, achieving much better data efficiency. Finally, we compare EquiformerV2 with Equiformer on QM9 dataset (Ramakrishnan et al., 2014; Ruddigkeit et al., 2012) and OC20 S2EF-2M dataset to better understand the performance gain of higher degrees and the improved architecture.

## 2 BACKGROUND

We present background relevant to this work here and discuss related works in Sec. B.

### 2.1 $SE(3)/E(3)$-EQUIVARIANT NEURAL NETWORKS

We discuss the relevant background of $SE(3)/E(3)$-equivariant neural networks here. Please refer to Sec. A in appendix for more details of equivariance and group theory.

Including equivariance in neural networks can serve as a strong prior knowledge, which can therefore improve data efficiency and generalization. Equivariant neural networks use equivariant irreps features built from vector spaces of irreducible representations (irreps) to achieve equivariance to 3D rotation. Specifically, the vector spaces are $(2L+1)$-dimensional, where degree $L$ is a non-negative integer. $L$ can be intuitively interpreted as the angular frequency of the vectors, i.e., how fast the vectors rotate with respect to a rotation of the coordinate system. Higher $L$ is critical to tasks sensitive to angular

information like predicting forces (Batzner et al., 2022; Zitnick et al., 2022; Passaro & Zitnick, 2023). Vectors of degree $L$ are referred to as type-$L$ vectors, and they are rotated with Wigner-D matrices $D^{(L)}$ when rotating coordinate systems. Euclidean vectors $\vec{r}$ in $\mathbb{R}^3$ can be projected into type-$L$ vectors by using spherical harmonics $Y^{(L)}(\frac{\vec{r}}{||\vec{r}||})$. We use order $m$ to index the elements of type-$L$ vectors, where $-L \leqslant m \leqslant L$. We concatenate multiple type-$L$ vectors to form an equivariant irreps feature $f$. Concretely, $f$ has $C_L$ type-$L$ vectors, where $0 \leqslant L \leqslant L_{max}$ and $C_L$ is the number of channels for type-$L$ vectors. In this work, we mainly consider $C_L = C$, and the size of $f$ is $(L_{max} + 1)^2 \times C$. We index $f$ by channel $i$, degree $L$, and order $m$ and denote as $f_{m,i}^{(L)}$.

Equivariant GNNs update irreps features by passing messages of transformed irreps features between nodes. To interact different type-$L$ vectors during message passing, we use tensor products, which generalize multiplication to equivariant irreps features. Denoted as $\otimes_{L_1, L_2}^{L_3}$, the tensor product uses Clebsch-Gordan coefficients to combine type-$L_1$ vector $f^{(L_1)}$ and type-$L_2$ vector $g^{(L_2)}$ and produces type-$L_3$ vector $h^{(L_3)}$:

$$h_{m_3}^{(L_3)} = (f^{(L_1)} \otimes_{L_1, L_2}^{L_3} g^{(L_2)})_{m_3} = \sum_{m_1=-L_1}^{L_1} \sum_{m_2=-L_2}^{L_2} C_{(L_1, m_1)(L_2, m_2)}^{(L_3, m_3)} f_{m_1}^{(L_1)} g_{m_2}^{(L_2)} \quad (1)$$

where $m_1$ denotes order and refers to the $m_1$-th element of $f^{(L_1)}$. Clebsch-Gordan coefficients $C_{(L_1, m_1)(L_2, m_2)}^{(L_3, m_3)}$ are non-zero only when $|L_1 - L_2| \leqslant L_3 \leqslant |L_1 + L_2|$ and thus restrict output vectors to be of certain degrees. We typically discard vectors with $L > L_{max}$, where $L_{max}$ is a hyper-parameter, to prevent vectors of increasingly higher dimensions. In many works, message passing is implemented as equivariant convolutions, which perform tensor products between input irreps features $x^{(L_1)}$ and spherical harmonics of relative position vectors $Y^{(L_2)}(\frac{\vec{r}}{||\vec{r}||})$.

## 2.2 EQUIFORMER

Equiformer (Liao & Smidt, 2023) is an $SE(3)/E(3)$-equivariant GNN that combines the inductive biases of equivariance with the strength of Transformers. First, Equiformer replaces scalar node features with equivariant irreps features to incorporate equivariance. Next, it performs equivariant operations on these irreps features and equivariant graph attention for message passing. These operations include tensor products and equivariant linear operations, equivariant layer normalization (Ba et al., 2016) and gate activation (Weiler et al., 2018). For stronger expressivity in the attention compared to typical Transformers, Equiformer uses non-linear functions for both attention weights and message passing. Additionally, Equiformer incorporates regularization techniques commonly used by Transformers, e.g., dropout (Srivastava et al., 2014) to attention weights (Veličković et al., 2018) and stochastic depth (Huang et al., 2016) to the outputs of equivariant graph attention and feed forward networks.

## 2.3 eSCN CONVOLUTION

eSCN convolutions (Passaro & Zitnick, 2023) are proposed to use $SO(2)$ linear operations for efficient tensor products. We provide an outline and intuition for their method here, please refer to Sec. A.3 and their work (Passaro & Zitnick, 2023) for mathematical details.

A traditional $SO(3)$ convolution interacts input irreps features $x_{m_i}^{(L_i)}$ and spherical harmonic projections of relative positions $Y_{m_f}^{(L_f)}(\vec{r}_{ij})$ with an $SO(3)$ tensor product with Clebsch-Gordan coefficients $C_{(L_i, m_i), (L_f, m_f)}^{(L_o, m_o)}$. The projection $Y_{m_f}^{(L_f)}(\vec{r}_{ij})$ becomes sparse if we rotate the relative position vector $\vec{r}_{ij}$ with a rotation matrix $D_{ij}$ to align with the direction of $L = 0$ and $m = 0$, which corresponds to the z axis traditionally but the y axis in the conventions of e3nn (Geiger et al., 2022). Concretely, given $D_{ij}\vec{r}_{ij}$ aligned with the y axis, $Y_{m_f}^{(L_f)}(D_{ij}\vec{r}_{ij}) \neq 0$ only for $m_f = 0$. If we consider only $m_f = 0$, $C_{(L_i, m_i), (L_f, m_f)}^{(L_o, m_o)}$ can be simplified, and $C_{(L_i, m_i), (L_f, 0)}^{(L_o, m_o)} \neq 0$ only when $m_i = \pm m_o$. Therefore, the original expression depending on $m_i$, $m_f$, and $m_o$ is now reduced to only depend on $m_o$. This means we are no longer mixing all integer values of $m_i$ and $m_f$, and outputs of order $m_o$ are linear combinations of inputs of order $\pm m_o$. eSCN convolutions go one step further and replace the remaining non-trivial paths of the $SO(3)$ tensor product with an $SO(2)$ linear operation to allow for additional parameters of interaction between $\pm m_o$ without breaking equivariance.

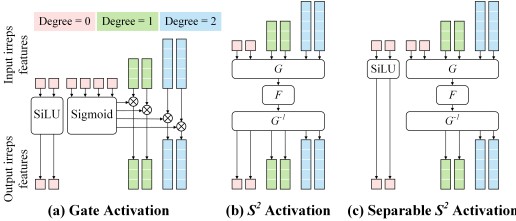 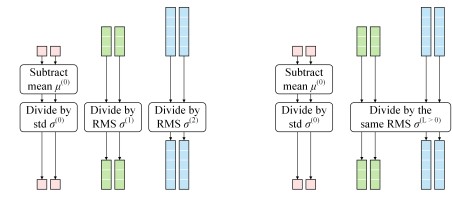

Figure 2: Illustration of different activation functions. $G$ denotes conversion from vectors to point samples on a sphere, $F$ can typically be a SiLU activation or MLPs, and $G^{-1}$ is the inverse of $G$.

Figure 3: Illustration of how statistics are calculated in different normalizations. "std" denotes standard deviation, and "RMS" denotes root mean square.

## 3 EQUIFORMERV2

Starting from Equiformer (Liao & Smidt, 2023), we first use eSCN convolutions to scale to higher-degree representations (Sec. 3.1). Then, we propose three architectural improvements, which yield further performance gain when using higher degrees: attention re-normalization (Sec. 3.2), separable $S^2$ activation (Sec. 3.3) and separable layer normalization (Sec. 3.4). Figure 1 illustrates the overall architecture of EquiformerV2 and the differences from Equiformer.

### 3.1 INCORPORATING eSCN CONVOLUTIONS FOR HIGHER DEGREES

The computational complexity of $SO(3)$ tensor products used in traditional $SO(3)$ convolutions scale unfavorably with $L_{max}$. Because of this, it is impractical for Equiformer to use $L_{max} > 2$ for large-scale datasets like OC20. Since higher $L_{max}$ can better capture angular information and are correlated with model expressivity (Batzner et al., 2022), low values of $L_{max}$ can lead to limited performance on certain tasks such as predicting forces. Therefore, we replace original tensor products with eSCN convolutions for efficient tensor products, enabling Equiformer to scale up $L_{max}$ to 6 or 8 on OC20 dataset. Equiformer uses equivariant graph attention for message passing. The attention consists of depth-wise tensor products, which mix information across different degrees, and linear layers, which mix information between channels of the same degree. Since eSCN convolutions mix information across both degrees and channels, we replace the $SO(3)$ convolution, which involves one depth-wise tensor product layer and one linear layer, with a single eSCN convolutional layer, which consists of a rotation matrix $D_{ij}$ and an $SO(2)$ linear layer as shown in Figure 1b.

### 3.2 ATTENTION RE-NORMALIZATION

Equivariant graph attention in Equiformer uses tensor products to project node embeddings $x_i$ and $x_j$, which contain vectors of degrees from 0 to $L_{max}$, to scalar features $f_{ij}^{(0)}$ and applies non-linear functions to $f_{ij}^{(0)}$ for attention weights $a_{ij}$. We propose attention re-normalization and introduce one additional layer normalization (LN) (Ba et al., 2016) before non-linear functions. Specifically, given $f_{ij}^{(0)}$, we first apply LN and then use one leaky ReLU layer and one linear layer to calculate $z_{ij} = w_a^\top \text{LeakyReLU}(\text{LN}(f_{ij}^{(0)}))$ and $a_{ij} = \text{softmax}_j(z_{ij}) = \frac{\exp(z_{ij})}{\sum_{k \in \mathcal{N}(i)} \exp(z_{ik})}$, where $w_a$ is a learnable vector of the same dimension as $f_{ij}^{(0)}$. The motivation is similar to ViT-22B (Dehghani et al., 2023), where they find that they need an additional layer normalization to stabilize training when increasing widths. When we scale up $L_{max}$, we effectively increase the number of input channels for calculating $f_{ij}^{(0)}$. By normalizing $f_{ij}^{(0)}$, the additional LN can make sure the inputs to subsequent non-linear functions and softmax operations still lie within the same range as lower $L_{max}$ is used. This empirically improves the performance as shown in Table 1a.

### 3.3 SEPARABLE $S^2$ ACTIVATION

The gate activation (Weiler et al., 2018) used by Equiformer applies sigmoid activation to scalar features to obtain non-linear weights and then multiply irreps features of degree $> 0$ with non-linear weights to add non-linearity to equivariant features. The activation only accounts for the interaction from vectors of degree 0 to those of degree $> 0$ and can be sub-optimal when we scale up $L_{max}$. To better mix the information across degrees, SCN (Zitnick et al., 2022) and eSCN adopt $S^2$ activation (Cohen et al., 2018). The activation first converts vectors of all degrees to point samples on a sphere for each channel, applies unconstrained functions $F$ to those samples, and finally convert them back to vectors. Specifically, given an input irreps feature $x \in \mathbb{R}^{(L_{max}+1)^2 \times C}$, the output is $y = G^{-1}(F(G(x)))$, where $G$ denotes the conversion from vectors to point samples on a sphere, $F$

can be typical SiLU activation (Elfwing et al., 2017; Ramachandran et al., 2017) or typical MLPs, and $G^{-1}$ is the inverse of $G$. We provide more details of $S^2$ activation in Sec. A.4.

However, we find that directly replacing the gate activation with $S^2$ activation in Equiformer results in large gradients and training instability (Index 3 in Table 1a). To address the issue, we propose separable $S^2$ activation, which separates activation for vectors of degree 0 and those of degree $> 0$. Similar to gate activation, we have more channels for vectors of degree 0. As shown in Figure 2c, we apply a SiLU activation to the first part of vectors of degree 0, and the second part of vectors of degree 0 are used for $S^2$ activation along with vectors of higher degrees. After $S^2$ activation, we concatenate the first part of vectors of degree 0 with vectors of degrees $> 0$ as the final output and ignore the second part of vectors of degree 0. Separating the activation for vectors of degree 0 and those of degree $> 0$ prevents large gradients, enabling using more expressive $S^2$ activation for better performance. Additionally, we use separable $S^2$ activation in feed forward networks (FFNs). Figure 2 illustrates the differences between gate activation, $S^2$ activation and separable $S^2$ activation.

## 3.4 SEPARABLE LAYER NORMALIZATION

Equivariant layer normalization used by Equiformer normalizes vectors of different degrees independently. However, it potentially ignores the relative importance of different degrees since the relative magnitudes between different degrees become the same after the normalization. Therefore, instead of performing normalization to each degree independently, motivated by the separable $S^2$ activation mentioned above, we propose separable layer normalization (SLN), which separates normalization for vectors of degree 0 and those of degrees $> 0$. Mathematically, let $x \in \mathbb{R}^{(L_{max}+1)^2 \times C}$ denote an input irreps feature of maximum degree $L_{max}$ and $C$ channels, and $x_{m,i}^{(L)}$ denote the $L$-th degree, $m$-th order and $i$-th channel of $x$. SLN calculates the output $y$ as follows. For $L = 0$, $y^{(0)} = \gamma^{(0)} \circ \left( \frac{x^{(0)} - \mu^{(0)}}{\sigma^{(0)}} \right) + \beta^{(0)}$, where $\mu^{(0)} = \frac{1}{C} \sum_{i=1}^{C} x_{0,i}^{(0)}$ and $\sigma^{(0)} = \sqrt{\frac{1}{C} \sum_{i=1}^{C} (x_{0,i}^{(0)} - \mu^{(0)})^2}$.

For $L > 0$, $y^{(L)} = \gamma^{(L)} \circ \left( \frac{x^{(L)}}{\sigma^{(L>0)}} \right)$, where $\sigma^{(L>0)} = \sqrt{\frac{1}{L_{max}} \sum_{L=1}^{L_{max}} \left( \sigma^{(L)} \right)^2}$ and $\sigma^{(L)} = \sqrt{\frac{1}{C} \sum_{i=1}^{C} \frac{1}{2L+1} \sum_{m=-L}^{L} \left( x_{m,i}^{(L)} \right)^2}$. $\gamma^{(0)}, \gamma^{(L)}, \beta^{(0)} \in \mathbb{R}^C$ are learnable parameters, $\mu^{(0)}$ and $\sigma^{(0)}$

are mean and standard deviation of vectors of degree 0, $\sigma^{(L)}$ and $\sigma^{(L>0)}$ are root mean square values (RMS), and $\circ$ denotes element-wise product. Figure 3 compares how $\mu^{(0)}, \sigma^{(0)}, \sigma^{(L)}$ and $\sigma^{(L>0)}$ are calculated in equivariant layer normalization and SLN. Preserving the relative magnitudes between degrees $> 0$ improves performance as shown in Table 1a.

## 3.5 OVERALL ARCHITECTURE

**Equivariant Graph Attention.** Figure 1b illustrates equivariant graph attention after the above modifications. Given node embeddings $x_i$ and $x_j$, we first concatenate them along the channel dimension and then rotate them with rotation matrices $D_{ij}$ based on their relative positions or edge directions $\vec{r}_{ij}$. We replace depth-wise tensor products and linear layers between $x_i$, $x_j$ and $f_{ij}$ with a single $SO(2)$ linear layer. To consider the information of relative distances $||\vec{r}_{ij}||$, we transform $||\vec{r}_{ij}||$ with a radial function to obtain edge distance embeddings and then multiply the edge distance embeddings with concatenated node embeddings before the first $SO(2)$ linear layer. We split the outputs $f_{ij}$ of the first $SO(2)$ linear layer into two parts. The first part is scalar features $f_{ij}^{(0)}$, which only contains vectors of degree 0, and the second part is irreps features $f_{ij}^{(L)}$ and includes vectors of all degrees up to $L_{max}$. As mentioned in Sec. 3.2, we first apply an additional LN to $f_{ij}^{(0)}$ and then follow the design of Equiformer by applying one leaky ReLU layer, one linear layer and a final softmax layer to obtain attention weights $a_{ij}$. As for value $v_{ij}$, we replace the gate activation with separable $S^2$ activation with $F$ being a single SiLU activation and then apply the second $SO(2)$ linear layer. While in Equiformer, the message $m_{ij}$ sent from node $j$ to node $i$ is $m_{ij} = a_{ij} \times v_{ij}$, here we need to rotate $a_{ij} \times v_{ij}$ back to original coordinate frames and the message $m_{ij}$ becomes $D_{ij}^{-1}(a_{ij} \times v_{ij})$. Finally, we can perform $h$ parallel equivariant graph attention functions given $f_{ij}$. The $h$ different outputs are concatenated and projected with a linear layer to become the final output $y_i$. Parallelizing attention functions and concatenating can be implemented with "Reshape".

**Feed Forward Network.** As shown in Figure 1d, we replace the gate activation with separable $S^2$ activation. The function $F$ consists of a two-layer MLP, with each linear layer followed by SiLU, and a final linear layer.

| Index | Attention re-normalization | Activation | Normalization | Epochs | forces | energy |
|---|---|---|---|---|---|---|
| 1 | ✗ | Gate | LN | 12 | 21.85 | 286 |
| 2 | ✓ | Gate | LN | 12 | 21.86 | 279 |
| 3 | ✓ | $S^2$ | LN | 12 | didn't converge | |
| 4 | ✓ | Sep. $S^2$ | LN | 12 | 20.77 | 285 |
| 5 | ✓ | Sep. $S^2$ | SLN | 12 | 20.46 | 285 |
| 6 | ✓ | Sep. $S^2$ | LN | 20 | 20.02 | 276 |
| 7 | ✓ | Sep. $S^2$ | SLN | 20 | 19.72 | 278 |
| 8 | eSCN baseline | | | 12 | 21.3 | 294 |

(a) Architectural improvements. Attention re-normalization improves energies, and separable $S^2$ activation ("Sep. $S^2$") and separable layer normalization ("SLN") improve forces.

| | | eSCN | | EquiformerV2 | |
|---|---|---|---|---|---|
| $L_{max}$ | Epochs | forces | energy | forces | energy |
| 6 | 12 | 21.3 | 294 | 20.46 | 285 |
| 6 | 20 | 20.6 | 290 | 19.78 | 280 |
| 6 | 30 | 20.1 | 285 | 19.42 | 278 |
| 8 | 12 | 21.3 | 296 | 20.46 | 279 |
| 8 | 20 | - | - | 19.95 | 273 |

(b) Training epochs. Training for more epochs consistently leads to better results.

| | eSCN | | EquiformerV2 | |
|---|---|---|---|---|
| $L_{max}$ | forces | energy | forces | energy |
| 4 | 22.2 | 291 | 21.37 | 284 |
| 6 | 21.3 | 294 | 20.46 | 285 |
| 8 | 21.3 | 296 | 20.46 | 279 |

(c) Degrees $L_{max}$. Higher degrees are consistently helpful.

| | eSCN | | EquiformerV2 | |
|---|---|---|---|---|
| $M_{max}$ | forces | energy | forces | energy |
| 2 | 21.3 | 294 | 20.46 | 285 |
| 3 | 21.2 | 295 | 20.24 | 284 |
| 4 | 21.2 | 298 | 20.24 | 282 |
| 6 | - | - | 20.26 | 278 |

(d) Orders $M_{max}$. Higher orders mainly improve energy predictions.

| | eSCN | | EquiformerV2 | |
|---|---|---|---|---|
| Layers | forces | energy | forces | energy |
| 8 | 22.4 | 306 | 21.18 | 293 |
| 12 | 21.3 | 294 | 20.46 | 285 |
| 16 | 20.5 | 283 | 20.11 | 282 |

(e) Number of Transformer blocks. Adding more blocks can help both force and energy predictions.

Table 1: Ablation studies of EquiformerV2. We report mean absolute errors (MAE) for forces in meV/Å and energy in meV, and lower is better. All models are trained on the 2M split of the OC20 S2EF dataset, and errors are averaged over the four validation sub-splits. The base model setting is marked in gray .

**Embedding.** This module consists of atom embedding and edge-degree embedding. The former is the same as that in Equiformer. For the latter, as depicted in the right branch in Figure 1c, we replace original linear layers and depth-wise tensor products with a single $SO(2)$ linear layer followed by a rotation matrix $D_{ij}^{-1}$. Similar to equivariant graph attention, we consider the information of relative distances by multiplying the outputs of the $SO(2)$ linear layer with edge distance embeddings.

**Radial Basis and Radial Function.** We represent relative distances $||\vec{r}_{ij}||$ with a finite radial basis like Gaussian radial basis functions (Schütt et al., 2017) to capture their subtle changes. We transform radial basis with a learnable radial function to generate edge distance embeddings. The function consists of a two-layer MLP, with each linear layer followed by LN and SiLU, and a final linear layer.

**Output Head.** To predict scalar quantities like energy, we use a feed forward network to transform irreps features on each node into a scalar and then sum over all nodes. For predicting atom-wise forces, we use a block of equivariant graph attention and treat the output of degree 1 as our predictions.

## 4 EXPERIMENTS

### 4.1 OC20 DATASET

Our main experiments focus on the large and diverse OC20 dataset (Chanussot* et al., 2021). Please refer to Sec. D.1 for details on the dataset. We first conduct ablation studies on EquiformerV2 trained on the 2M split of OC20 S2EF dataset (Sec. 4.1.1). Then, we report the results of training All and All+MD splits (Sec. 4.1.2). Additionally, we investigate the performance of EquiformerV2 when used in the AdsorbML algorithm (Lan et al., 2022) (Sec. 4.1.3). Please refer to Sec. C and D for details of models and training.

### 4.1.1 ABLATION STUDIES

**Architectural Improvements.** In Table 1a, we start with incorporating eSCN convolutions into Equiformer for higher-degree representations (Index 1) and then ablate the three proposed architectural improvements. First, with attention re-normalization, energy mean absolute errors (MAE) improve by 2.4%, while force MAE are about the same (Index 1 and 2). Second, we replace the gate activation with $S^2$ activation, but that does not converge (Index 3). With the proposed $S^2$ activation, we stabilize training and successfully leverage more expressive activation to improve force MAE (Index 4). Third, replacing equivariant layer normalization with separable layer normalization further improves force MAE (Index 5). We note that simply incorporating eSCN convolutions into Equiformer and using higher degrees (Index 1) do not result in better performance than the original eSCN baseline (Index 8), and that the proposed architectural changes are necessary. We additionally compare the performance gain of architectural improvements with that of training for longer. Attention re-normalization (Index 1 and 2) improves energy MAE by the same amount as increasing training epochs from 12 to 20

| Training set | Model | Number of parameters ↓ | Training time (GPU-days) ↓ | Throughput Samples / GPU sec. ↑ | S2EF validation Energy MAE (meV) ↓ | S2EF validation Force MAE (meV/Å) ↓ | S2EF test Energy MAE (meV) ↓ | S2EF test Force MAE (meV/Å) ↓ | IS2RS test AFbT (%) ↑ | IS2RS test ADwT (%) ↑ | IS2RE test Energy MAE (meV) ↓ |
|---|---|---|---|---|---|---|---|---|---|---|---|
| OC20 S2EF-All | SchNet (Schütt et al., 2017) | 9.1M | 194 | - | 549 | 56.8 | 540 | 54.7 | - | 14.4 | 749 |
| | DimeNet++-L-F+E (Gasteiger et al., 2020a) | 10.7M | 1600 | 4.6 | 515 | 32.8 | 480 | 31.3 | 21.7 | 51.7 | 559 |
| | SpinConv (Shuaibi et al., 2021) | 8.5M | 275 | 6.0 | 371 | 41.2 | 336 | 29.7 | 16.7 | 53.6 | 437 |
| | GemNet-dT (Klicpera et al., 2021) | 32M | 492 | 25.8 | 315 | 27.2 | 292 | 24.2 | 27.6 | 58.7 | 400 |
| | GemNet-OC (Gasteiger et al., 2022) | 39M | 336 | 18.3 | 244 | 21.7 | 233 | 20.7 | 35.3 | 60.3 | 355 |
| | SCN L=8 K=20 (Zitnick et al., 2022) | 271M | 645 | - | - | - | 244 | 17.7 | 40.3 | 67.1 | 330 |
| | eSCN L=6 K=20 (Passaro & Zitnick, 2023) | 200M | 600 | 2.9 | - | - | 242 | 17.1 | 48.5 | 65.7 | 341 |
| | EquiformerV2 ($\lambda_E = 2$, 153M) | 153M | 1368 | 1.8 | **236** | **15.7** | 229 | 14.8 | **53.0** | **69.0** | 316 |
| OC20 S2EF-All+MD | GemNet-OC-L-E (Gasteiger et al., 2022) | 56M | 640 | 7.5 | 239 | 22.1 | 230 | 21.0 | - | - | - |
| | GemNet-OC-L-F (Gasteiger et al., 2022) | 216M | 765 | 3.2 | 252 | 20.0 | 241 | 19.0 | 40.6 | 60.4 | - |
| | GemNet-OC-L-F+E (Gasteiger et al., 2022) | - | - | - | - | - | - | - | - | - | 348 |
| | SCN L=6 K=16 (4-tap 2-band) (Zitnick et al., 2022) | 168M | 414 | - | - | - | 228 | 17.8 | 43.3 | 64.9 | 328 |
| | SCN L=8 K=20 (Zitnick et al., 2022) | 271M | 1280 | - | - | - | 237 | 17.2 | 43.6 | 67.5 | 321 |
| | eSCN L=6 K=20 (Passaro & Zitnick, 2023) | 200M | 568 | 2.9 | 243 | 17.1 | 228 | 15.6 | 50.3 | 66.7 | 323 |
| | EquiformerV2 ($\lambda_E = 4$, 31M) | 31M | 705 | 7.1 | 232 | 16.3 | 228 | 15.5 | 47.6 | 68.3 | 315 |
| | EquiformerV2 ($\lambda_E = 2$, 153M) | 153M | 1368 | 1.8 | **230** | **14.6** | 227 | **13.8** | **55.4** | **69.8** | **311** |
| | EquiformerV2 ($\lambda_E = 4$, 153M) | 153M | 1571 | 1.8 | **227** | 15.0 | **219** | 14.2 | 54.4 | 69.4 | **309** |

Table 2: OC20 results on S2EF validation and test splits, and IS2RS and IS2RE test splits when trained on OC20 S2EF-All or S2EF-All+MD splits. Throughput is reported as the number of structures processed per GPU-second during training and measured on V100 GPUs. $\lambda_E$ is the coefficient of the energy loss.

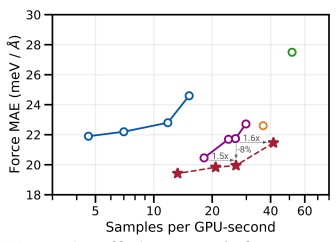 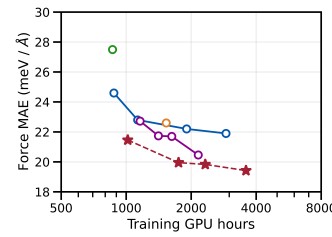 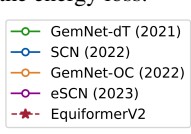

(a) Trade-offs between inference speed and validation force MAE.

(b) Trade-offs between training cost and validation force MAE.

Figure 4: EquiformerV2 achieves better accuracy trade-offs both in terms of inference speed as well as training cost. All models are trained on the S2EF-2M split and measured on V100 GPUs.

(Index 5 and 7). The improvement of SLN (Index 4 and 5) in force MAE is about $40\%$ of that of increasing training epochs from 12 to 20 (Index 5 and 7), and SLN is about $6\%$ faster in training. We also conduct the ablation studies on the QM9 dataset and summarize the results in Sec. F.3.

**Scaling of Parameters.** In Tables 1c, 1d, 1e, we vary the maximum degree $L_{max}$, the maximum order $M_{max}$, and the number of Transformer blocks and compare with equivalent eSCN variants. Across all experiments, EquiformerV2 performs better than its eSCN counterparts. Besides, while one might intuitively expect higher resolutions and larger models to perform better, this is only true for EquiformerV2, not eSCN. For example, increasing $L_{max}$ from 6 to 8 or $M_{max}$ from 3 to 4 degrades the performance of eSCN on energy predictions but helps that of EquiformerV2. In Table 1b, we show that increasing the training epochs from 12 to 30 epochs can consistently improve performance.

**Speed-Accuracy Trade-offs.** We compare trade-offs between inference speed or training time and force MAE among prior works and EquiformerV2 and summarize the results in Figure 4a and Figure 4b. EquifomerrV2 achieves the lowest force MAE across a wide range of inference speed and training cost.

### 4.1.2 MAIN RESULTS

Table 2 reports results on the test splits for all the three tasks of OC20 averaged over all four sub-splits. Models are trained on either OC20 S2EF-All or S2EF-All+MD splits. All test results are computed via the EvalAI evaluation server[1]. We train EquiformerV2 of two sizes, one with 153M parameters and the other with 31M parameters. When trained on the S2EF-All+MD split, EquiformerV2 ($\lambda_E = 4$, 153M) improves previous state-of-the-art S2EF energy MAE by $4\%$, S2EF force MAE by $9\%$, IS2RS Average Forces below Threshold (AFbT) by absolute $4\%$ and IS2RE energy MAE by $4\%$. In particular, the improvement in force predictions is significant. Going from SCN to eSCN, S2EF test force MAE improves from 17.2 meV/Å to 15.6 meV/Å , largely due to replacing approximate equivariance in SCN with strict equivariance in eSCN during message passing. Similarly, by scaling up the degrees of representations in Equiformer, EquiformerV2 ($\lambda_E = 4$, 153M) further improves force MAE to 14.2 meV/Å, which is similar to the gain of going from SCN to eSCN. Additionally, the smaller EquiformerV2 ($\lambda_E = 4$, 31M) improves upon previously best results for all metrics except IS2RS AFbT and achieves comparable training throughput to the fastest GemNet-OC-L-E. Although the training time of EquiformerV2 is higher here, we note that this is because training EquiformerV2

---

[1] eval.ai/web/challenges/challenge-page/712

| Model | GPU-seconds per relaxation ↓ | k = 1 Success | k = 1 Speedup | k = 2 Success | k = 2 Speedup | k = 3 Success | k = 3 Speedup | k = 4 Success | k = 4 Speedup | k = 5 Success | k = 5 Speedup |
|---|---|---|---|---|---|---|---|---|---|---|---|
| SchNet (Schütt et al., 2017) | - | 2.77% | 4266.13 | 3.91% | 2155.36 | 4.32% | 1458.77 | 4.73% | 1104.88 | 5.04% | 892.79 |
| DimeNet++ (Gasteiger et al., 2020a) | - | 5.34% | 4271.23 | 7.61% | 2149.78 | 8.84% | 1435.21 | 10.07% | 1081.96 | 10.79% | 865.20 |
| PaiNN (Schütt et al., 2021) | - | 27.44% | 4089.77 | 33.61% | 2077.65 | 36.69% | 1395.55 | 38.64% | 1048.63 | 39.57% | 840.44 |
| GemNet-OC (Gasteiger et al., 2022) | 9.4 | 68.76% | 4185.18 | 77.29% | 2087.11 | 80.78% | 1392.51 | 81.50% | 1046.85 | 82.94% | 840.25 |
| GemNet-OC-MD (Gasteiger et al., 2022) | 9.4 | 68.76% | 4182.04 | 78.21% | 2092.27 | 81.81% | 1404.11 | 83.25% | 1053.36 | 84.38% | 841.64 |
| GemNet-OC-MD-Large (Gasteiger et al., 2022) | 45.0 | 73.18% | 4078.76 | 79.65% | 2065.15 | 83.25% | 1381.39 | 85.41% | 1041.50 | 86.02% | 834.46 |
| SCN-MD-Large (Zitnick et al., 2022) | 120.0 | 77.80% | 3974.22 | 84.28% | 1989.32 | 86.33% | 1331.43 | 87.36% | 1004.40 | 87.77% | 807.00 |
| EquiformerV2 ($\lambda_E = 4$, 31M) | 12.2 | 84.17% | 3983.41 | 87.15% | 1992.64 | 87.87% | 1331.35 | 88.69% | 1000.62 | 89.31% | 802.95 |
| EquiformerV2 ($\lambda_E = 4$, 153M) | 45.0 | **85.41%** | 4001.71 | **88.90%** | 2012.47 | **90.54%** | 1352.08 | **91.06%** | 1016.31 | **91.57%** | 815.87 |

Table 3: AdsorbML results with EquiformerV2 ($\lambda_E = 4$, 31M) and ($\lambda_E = 4$, 153M) trained on S2EF-All+MD from Table 2. We visualize the speed-accuracy trade-offs of different models in Figure 5.

| Model | Training Set | Linear reference | Number of parameters | S2EF-Total validation Energy MAE (meV) ↓ ID | OOD | Force MAE (meV/Å) ↓ ID | OOD | S2EF-Total test Energy MAE (meV) ↓ ID | OOD | Force MAE (meV/Å) ↓ ID | OOD | IS2RE-Total test Energy MAE (meV) ↓ ID | OOD |
|---|---|---|---|---|---|---|---|---|---|---|---|---|---|
| GemNet-OC (Gasteiger et al., 2022) | OC22 | | 39M | 545 | 1011 | 30 | 40 | 374 | 829 | 29.4 | 39.6 | 1329 | 1584 |
| GemNet-OC (Gasteiger et al., 2022) | OC22 | ✓ | 39M | - | - | - | - | 357 | 1057 | 30.0 | 40.0 | - | - |
| GemNet-OC (Gasteiger et al., 2022) | OC22 + OC20 | ✓ | 39M | 464 | 859 | 27 | 34 | 311 | 689 | 26.9 | 34.2 | 1200 | 1534 |
| eSCN (Passaro & Zitnick, 2023) | OC22 | ✓ | 200M | - | - | - | - | 350 | 789 | 23.8 | 35.7 | - | - |
| EquiformerV2 ($\lambda_E = 1$, $\lambda_F = 1$) | OC22 | ✓ | 122M | **343** | **580** | 24.42 | 34.31 | **182.8** | 677.4 | 22.98 | 35.57 | **1084** | 1444 |
| EquiformerV2 ($\lambda_E = 4$, $\lambda_F = 100$) | OC22 | ✓ | 122M | 433 | 629 | **22.88** | **30.70** | 263.7 | **659.8** | **21.58** | **32.65** | 1119 | **1440** |

Table 4: OC22 results on S2EF-Total validation and test splits and IS2RE-Total test split.

for longer keeps improving performance and that we already demonstrate EquiformerV2 achieves better trade-offs between force MAE and speed.

### 4.1.3 ADSORBML RESULTS

Lan et al. (2022) recently proposes the AdsorbML algorithm, wherein they show that recent state-of-the-art GNNs can achieve more than $1000\times$ speedup over DFT relaxations at computing adsorption energies within a $0.1\text{eV}$ margin of DFT results with an $87\%$ success rate. This is done by using OC20-trained models to perform structure relaxations for an average 90 configurations of an adsorbate placed on a catalyst surface, followed by DFT single-point calculations for the top-$k$ structures with lowest predicted relaxed energies, as a proxy for calculating the global energy minimum or adsorption energy. We refer readers to Sec. D.4 and the work (Lan et al., 2022) for more details. We benchmark AdsorbML with EquiformerV2, and Table 3 shows that EquiformerV2 ($\lambda_E = 4$, 153M) improves over SCN by a significant margin, with $8\%$ and $5\%$ absolute improvements at $k = 1$ and $k = 2$, respectively. Moreover, EquiformerV2 ($\lambda_E = 4$, 153M) at $k = 2$ is more accurate at adsorption energy calculations than all the other models even at $k = 5$, thus requiring at least $2\times$ fewer DFT calculations. Since the speedup is with respect to using DFT for structure relaxations and that ML models are much faster than DFT, the speedup is dominated by the final DFT single-point calculations and ML models with the same value of $k$ have roughly the same speedup. To better understand the speed-accuracy trade-offs of different models, we also report average GPU-seconds of running one structure relaxation in Table 3. Particularly, EquiformerV2 ($\lambda_E = 4$, 31M) improves upon previous methods while being $3.7\times$ to $9.8\times$ faster than GemNet-OC-MD-Large and SCN, respectively.

### 4.2 OC22 DATASET

**Dataset.** The OC22 dataset (Tran* et al., 2022) focuses on oxide electrocatalysis. One crucial difference between OC22 and OC20 is that the energies in OC22 are DFT total energies. DFT total energies are harder to predict but are the most general and closest to a DFT surrogate, offering the flexibility to study property prediction beyond adsorption energies. Similar to OC20, the tasks in OC22 are S2EF-Total and IS2RE-Total. We train models on the OC22 S2EF-Total dataset and evaluate them on energy and force MAE on the S2EF-Total validation and test splits. We use the trained models to perform structural relaxations and predict relaxed energy. Relaxed energy predictions are evaluated on the IS2RE-Total test split.

**Training Details.** Please refer to Section E.1 for details on architectures, hyper-parameters and training time.

**Results.** We train two EquiformerV2 models with different energy coefficients $\lambda_E$ and force coefficients $\lambda_F$. We follow the practice of OC22 models and use linear reference (Tran* et al., 2022). The results are summarized in Table 4. EquiformerV2 improves upon previous models on all the tasks. EquiformerV2 ($\lambda_E = 4$, $\lambda_F = 100$) trained on only OC22 achieves better results on all the tasks than GemNet-OC trained on both OC20 and OC22. We note that OC22 contains about $8.4\text{M}$ structures and OC20 contains about $130\text{M}$ structures, and therefore EquiformerV2 demonstrates significantly better data efficiency. Additionally, the performance gap between eSCN and EquiformerV2 is larger than that on OC20, suggesting that more complicated structures can benefit more from the proposed architecture. When trained on OC20 S2EF-All+MD, EquiformerV2 ($\lambda_E = 4$, 153M) improves upon eSCN by $4\%$ on energy MAE and $9\%$ on force MAE. For OC22, EquiformerV2 ($\lambda_E = 4$, $\lambda_F = 100$) improves upon eSCN by $18.9\%$ on average energy MAE and $8.9\%$ on average force MAE.

| Model | Task Units | $\alpha$ $a_0^3$ | $\Delta\varepsilon$ meV | $\varepsilon_{\text{HOMO}}$ meV | $\varepsilon_{\text{LUMO}}$ meV | $\mu$ D | $C_\nu$ cal/mol K | $G$ meV | $H$ meV | $R^2$ $a_0^2$ | $U$ meV | $U_0$ meV | ZPVE meV |
|---|---|---|---|---|---|---|---|---|---|---|---|---|---|
| DimeNet++ (Gasteiger et al., 2020a) | | **.044** | 33 | 25 | 20 | .030 | .023 | 8 | 7 | .331 | 6 | 6 | 1.21 |
| EGNN (Satorras et al., 2021)[†] | | .071 | 48 | 29 | 25 | .029 | .031 | 12 | 12 | .106 | 12 | 11 | 1.55 |
| PaiNN (Schütt et al., 2021) | | .045 | 46 | 28 | 20 | .012 | .024 | **7.35** | **5.98** | .066 | **5.83** | **5.85** | 1.28 |
| TorchMD-NET (Thölke & Fabritiis, 2022) | | .059 | 36 | 20 | 18 | .011 | .026 | 7.62 | 6.16 | **.033** | 6.38 | 6.15 | 1.84 |
| SphereNet (Liu et al., 2022) | | .046 | 32 | 23 | 18 | .026 | **.021** | 8 | 6 | .292 | 7 | 6 | **1.12** |
| SEGNN (Brandstetter et al., 2022)[†] | | .060 | 42 | 24 | 21 | .023 | .031 | 15 | 16 | .660 | 13 | 15 | 1.62 |
| EQGAT (Le et al., 2022) | | .053 | 32 | 20 | 16 | .011 | .024 | 23 | 24 | .382 | 25 | 25 | 2.00 |
| Equiformer (Liao & Smidt, 2023) | | .046 | 30 | 15 | 14 | .011 | .023 | 7.63 | 6.63 | .251 | 6.74 | 6.59 | 1.26 |
| EquiformerV2 | | .050 | **29** | **14** | **13** | **.010** | .023 | 7.57 | 6.22 | .186 | 6.49 | 6.17 | 1.47 |

Table 5: Mean absolute error results on QM9 test set. † denotes using different data partitions.

| Model | $L_{max}$ | Energy MAE (meV)↓ | Force MAE (meV/Å)↓ | Training time (GPU-hours)↓ |
|---|---|---|---|---|
| Equiformer (Liao & Smidt, 2023) | 2 | 297 | 27.57 | 1365 |
| Equiformer (Liao & Smidt, 2023) | 3 | OOM | OOM | OOM |
| EquiformerV2 | 2 | 298 | 26.24 | 600 |
| EquiformerV2 | 4 | 284 | 21.37 | 966 |
| EquiformerV2 | 6 | 285 | 20.46 | 1412 |

Table 6: Comparison on OC20 S2EF-2M dataset. Errors are averaged over the four validation sub-splits. "OOM" denotes out-of-memory error, and we cannot use $L_{max} = 3$ for Equiformer.

### 4.3 COMPARISON WITH EQUIFORMER

**QM9 Dataset.** We follow the setting of Equiformer and train EquiformerV2 on the QM9 dataset (Ramakrishnan et al., 2014; Ruddigkeit et al., 2012) and summarize the results in Table 5. The performance gain of using higher degrees and the improved architecture is not as significant as that on OC20 and OC22 datasets. This, however, is not surprising. The training set of QM9 contains only 110k examples, which is much less than OC20 S2EF-2M with 2M examples and OC22 with 8.4M examples. Moreover, QM9 has much less numbers of atoms in each example and much less diverse atom types, and each example has less angular variations. Nevertheless, EquiformerV2 achieves better results than Equiformer on 9 out of the 12 tasks and is therefore the overall best performing model. We additionally train EquiformerV2 with Noisy Nodes (Godwin et al., 2022) to better understand the gain of higher degrees in Sec. F.1.

**OC20 S2EF-2M Dataset.** We use similar configurations (e.g., numbers of blocks and numbers of channels) and train Equiformer on OC20 S2EF-2M dataset for the same number of epochs as training EquiformerV2. We vary $L_{max}$ for both Equiformer and EquiformerV2 and compare the results in Table 6. For $L_{max} = 2$, EquiformerV2 is 2.3× faster than Equiformer since EquiformerV2 uses eSCN convolutions for efficient $SO(3)$ convolutions. Additionally, EquiformerV2 achieves better force MAE and similar energy MAE, demonstrating the effectiveness of the proposed improved architecture. For $L_{max} > 2$, we encounter out-of-memory errors when training Equiformer even after we reduce the number of blocks and use the batch size $= 1$. In contrast, We can easily train EquiformerV2 with $L_{max}$ up to 6. When increasing $L_{max}$ from 2 to 4, EquiformerV2 achieves lower energy MAE and significantly lower force MAE than Equiformer and requires 1.4× less training time. The comparison suggests that complicated datasets have more performance to gain from using more expressive models, enabling better performance and lower computational cost.

**Discussion.** One limitation of EquiformerV2 is that the performance gains brought by scaling to higher degrees and the proposed architectural improvements can depend on datasets and tasks. For small datasets like QM9, the performance gain is not significant. We additionally compare Equiformer and EquiformerV2 on OC20 IS2RE dataset in Sec. D.5. For different tasks, the improvements are also different, and force predictions benefit more from better expressivity than energy predictions. We note that the first issue can be mitigated by first pre-training on large datasets like OC20 and PCQM4Mv2 (Nakata & Shimazaki, 2017) optionally via denoising (Godwin et al., 2022; Zaidi et al., 2023) and then fine-tuning on smaller datasets. The second issue might be mitigated by combining DFT with ML models. For example, AdsorbML uses ML forces for structural relaxations and a single-point DFT for calculating the final relaxed energies.

## 5 CONCLUSION

In this work, we investigate how equivariant Transformers can be scaled up to higher degrees of equivariant representations. We start by replacing $SO(3)$ convolutions in Equiformer with eSCN convolutions, and then we propose three architectural improvements to better leverage the power of higher degrees – attention re-normalization, separable $S^2$ activation and separable layer normalization. With these modifications, we propose EquiformerV2, which outperforms state-of-the-art methods on all the tasks on the OC20 and OC22 datasets, improves speed-accuracy trade-offs, and achieves the best success rate when used in AdsorbML. We also compare EquiformerV2 with Equiformer to better understand the performance gain brought by higher degrees and the improved architecture.

## 6 ETHICS STATEMENT

EquiformerV2 achieves more accurate approximation of quantum mechanical calculations and demonstrates one further step toward being able to replace DFT compute force fields with machine learned ones for practical applications in chemistry and material science. We hope these promising results will encourage the community to make further progress in applications like material design and drug discovery, rather than use these methods for adversarial purposes. We note that these methods only facilitate the identification of molecules or materials with specific properties; there remain substantial hurdles to synthesize and deploy such molecules or materials at scale. Finally, we note that the proposed method is general and can be applied to different problems like protein structure prediction (Lee et al., 2022) as long as inputs can be modeled as 3D graphs.

## 7 REPRODUCIBILITY STATEMENT

We include details on architectures, hyper-parameters and training time in Sec. D.2 (OC20), Sec. E.1 (OC22) and Sec. F.2 (QM9).

The code for reproducing the results of EquiformerV2 trained on OC20 S2EF-2M and QM9 datasets is available at https://github.com/atomicarchitects/equiformer_v2.

## ACKNOWLEDGEMENT

We thank Larry Zitnick and Saro Passaro for helpful discussions. We also thank Muhammed Shuaibi for helping with the DFT evaluations for AdsorbML (Lan et al., 2022). We acknowledge the MIT SuperCloud and Lincoln Laboratory Supercomputing Center (Reuther et al., 2018) for providing high performance computing and consultation resources that have contributed to the research results reported within this paper.

Yi-Lun Liao and Tess Smidt were supported by DOE ICDI grant DE-SC0022215.

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

APPENDIX

## A    ADDITIONAL BACKGROUND

We first provide relevant mathematical background on group theory and equivariance. We note that most of the content is adapted from Equiformer (Liao & Smidt, 2023) and that these works (Zee, 2016; Dresselhaus et al., 2007) have more in-depth and pedagogical discussions. Then, we provide mathematical details of eSCN convolutions.

### A.1    GROUP THEORY

**Definition of Groups.**    A group is an algebraic structure that consists of a set $G$ and a binary operator $\circ : G \times G \to G$. Typically denoted as $G$, groups satisfy the following four axioms:

1. Closure: $g \circ h \in G$ for all $g, h \in G$.
2. Identity: There exists an identity element $e \in G$ such that $g \circ e = e \circ g = g$ for all $g \in G$.
3. Inverse: For each $g \in G$, there exists an inverse element $g^{-1} \in G$ such that $g \circ g^{-1} = g^{-1} \circ g = e$.
4. Associativity: $g \circ h \circ i = (g \circ h) \circ i = g \circ (h \circ i)$ for all $g, h, i \in G$.

In this work, we consider 3D Euclidean symmetry, and relevant groups are:

1. The Euclidean group in three dimensions $E(3)$: 3D rotation, translation and inversion.
2. The special Euclidean group in three dimensions $SE(3)$: 3D rotation and translation.
3. The orthogonal group in three dimensions $O(3)$: 3D rotation and inversion.
4. The special orthogonal group in three dimensions $SO(3)$: 3D rotation.

Since eSCN (Passaro & Zitnick, 2023) and this work only consider equivariance to 3D rotation and invariance to 3D translation but not inversion, we mainly discuss $SE(3)$-equivariance in the main text and in appendix and note that more details of $E(3)$-equivariance can be found in the work of Equiformer (Liao & Smidt, 2023).

**Group Representations.** Given a vector space $X$, the way a group $G$ acts on $X$ is given by the group representation $D_X$. $D_X$ is parameterized by $g \in G$, with $D_X(g) : X \to X$. Group representations $D_X$ are invertible matrices, and group transformations, or group actions, take the form of matrix multiplications. This definition of group representations satisfies the requirements of groups, including associativity, $D(g)D(h) = D(g \circ h)$ for all $g, h \in G$. We say that the two group representations $D(g)$ and $D'(g)$ are equivalent if there exists a change-of-basis $N \times N$ matrix $P$ such that $P^{-1}D(g)P = D'(g)$ for all $g \in G$. $D(g)$ is reducible if $D'(g)$ is block diagonal for all $g \in G$, meaning that $D'(g)$ acts on multiple independent subspaces of the vector space. Otherwise, the representation $D(g)$ is said to be irreducible. Irreducible representations, or irreps, are a class of representations that are convenient for composing different group representations. Specifically, for the case of $SO(3)$, Wigner-D matrices are irreducible representations, and we can express any group representation of $SO(3)$ as a direct sum (concatentation) of Wigner-D matrices (Zee, 2016; Dresselhaus et al., 2007; Geiger et al., 2022):

$$D(g) = P^{-1}\left(\bigoplus_i D^{(L_i)}(g)\right)P = P^{-1}\begin{pmatrix} D^{(L_0)}(g) & & \\ & D^{(L_1)}(g) & \\ & & \ddots \end{pmatrix}P \quad (2)$$

where $D^{(L_i)}(g)$ are Wigner-D matrices of degree $L_i$.

## A.2 EQUIVARIANCE

A function $f$ mapping between vector spaces $X$ and $Y$ is equivariant to a group of transformations $G$ if for any input $x \in X$, output $y \in Y$ and group element $g \in G$, we have $f(D_X(g)x) = D_Y(g)f(x) = D_Y(g)y$, where $D_X(g)$ and $D_Y(g)$ are transformation matrices or group representations parametrized by $g$ in $X$ and $Y$. Additionally, $f$ is invariant when $D_Y(g)$ is an identity matrix for any $g \in G$.

As neural networks comprise many composable operations, equivariant neural networks comprise many equivariant operations to maintain the equivariance of input, intermediate, and output features. Incorporating equivariance as a strong prior knowledge can improve data efficiency and generalization of neural networks (Batzner et al., 2022; Rackers et al., 2023; Frey et al., 2022). In this work, we achieve equivariance to 3D rotation by operating on vector spaces of $SO(3)$ irreps, incorporate invariance to 3D translation by acting on relative positions, but do not consider inversion.

## A.3 eSCN CONVOLUTION

Message passing is used to update equivariant irreps features and is typically implemented as $SO(3)$ convolutions. A traditional $SO(3)$ convolution interacts input irrep features $x_{m_i}^{(L_i)}$ and spherical harmonic projections of relative positions $Y_{m_f}^{(L_f)}(\vec{r}_{ts})$ with an $SO(3)$ tensor product with Clebsch-Gordan coefficients $C_{(L_i,m_i),(L_f,m_f)}^{(L_o,m_o)}$. Since tensor products are compute-intensive, eSCN convolutions (Passaro & Zitnick, 2023) are proposed to reduce the complexity of tensor products when they are used in $SO(3)$ convolutions. Rotating the input irreps features $x_{m_i}^{(L_i)}$ based on the relative position vectors $\vec{r}_{ts}$ simplifies the tensor products and enables reducing $SO(3)$ convolutions to $SO(2)$ linear operations. Below we provide the mathematical details of $SO(3)$ convolutions built from tensor products and how rotation can reduce their computational complexity.

Tensor products interact type-$L_i$ vector $x^{(L_i)}$ and type-$L_f$ vector $f^{(L_f)}$ to produce type-$L_o$ vector $y^{(L_o)}$ with Clebsch-Gordan coefficients $C_{(L_i,m_i),(L_f,m_f)}^{(L_o,m_o)}$. Clebsch-Gordan coefficients $C_{(L_i,m_i),(L_f,m_f)}^{(L_o,m_o)}$ are non-zero only when $|L_i - L_o| \leqslant L_f \leqslant |L_i + L_o|$. Each non-trivial combination of $L_i \otimes L_f \to L_o$ is called a path, and each path is independently equivariant and can be assigned a learnable weight $w_{L_i,L_f,L_o}$.

We consider the message $m_{ts}$ sent from source node $s$ to target node $t$ in an $SO(3)$ convolution. The $L_o$-th degree of $m_{ts}$ can be expressed as:

$$m_{ts}^{(L_o)} = \sum_{L_i,L_f} w_{L_i,L_f,L_o}\left(x_s^{(L_i)} \otimes_{L_i,L_f}^{L_o} Y^{(L_f)}(\hat{r}_{ts})\right) \quad (3)$$

where $x_s$ is the irreps feature at source node $s$, $x_s^{(L_i)}$ denotes the $L_i$-th degree of $x_s$, and $\hat{r}_{ts} = \frac{\vec{r}_{ts}}{|\vec{r}_{ts}|}$. The spherical harmonic projection of relative positions $Y^{(L_f)}(\hat{r}_{ts})$ becomes sparse if we rotate $\hat{r}_{ts}$ with a rotation matrix $R_{ts}$ to align with the direction of $L = 0$ and $m = 0$, which corresponds to the z axis traditionally but the y axis in the conventions of e3nn (Geiger et al., 2022). Concretely, given $R_{ts}\hat{r}_{ts}$ aligned with the y axis, $Y_{m_f}^{(L_f)}(R_{ts}\hat{r}_{ts}) \neq 0$ only for $m_f = 0$. Without loss of equivariance, we re-scale $Y_0^{(L_f)}(R_{ts}\vec{r}_{ts})$ to be one. Besides, we denote $D^{(L_i)}(R_{ts}) = D^{(L_i)}$ and $D^{(L_o)}(R_{ts}) = D^{(L_o)}$ as Wigner-D matrices of degrees $L_i$ and $L_o$ based on rotation matrix $R_{ts}$, respectively, and we define $D^{(L_i)}x_s^{(L_i)} = \tilde{x}_s^{(L_i)}$. Therefore, by rotating $x_s^{(L_i)}$ and $Y^{(L_f)}$ based on $\hat{r}_{ts}$, we can simplify Eq. 3 as follows:

$$m_{ts}^{(L_o)} = \left(D^{(L_o)}(R_{ts})\right)^{-1} \sum_{L_i,L_f} w_{L_i,L_f,L_o} \left(D^{(L_i)}(R_{ts})x_s^{(L_i)} \otimes_{L_i,L_f}^{L_o} Y^{(L_f)}(R_{ts}\hat{r}_{ts})\right)$$

$$= \left(D^{(L_o)}\right)^{-1} \sum_{L_i,L_f} w_{L_i,L_f,L_o} \bigoplus_{m_o} \left(\sum_{m_i,m_f} \left(D^{(L_i)}x_s^{(L_i)}\right)_{m_i} C_{(L_i,m_i),(L_f,m_f)}^{(L_o,m_o)} \left(Y^{(L_f)}(R_{ts}\hat{r}_{ts})\right)_{m_f}\right)$$

$$= \left(D^{(L_o)}\right)^{-1} \sum_{L_i,L_f} w_{L_i,L_f,L_o} \bigoplus_{m_o} \left(\sum_{m_i} \left(D^{(L_i)}x_s^{(L_i)}\right)_{m_i} C_{(L_i,m_i),(L_f,0)}^{(L_o,m_o)}\right)$$

$$= \left(D^{(L_o)}\right)^{-1} \sum_{L_i,L_f} w_{L_i,L_f,L_o} \bigoplus_{m_o} \left(\sum_{m_i} \left(\tilde{x}_s^{(L_i)}\right)_{m_i} C_{(L_i,m_i),(L_f,0)}^{(L_o,m_o)}\right)$$

$$(4)$$

where $\bigoplus$ denotes concatenation. Additionally, given $m_f = 0$, Clebsch-Gordan coefficients $C_{(L_i,m_i),(L_f,0)}^{(L_o,m_o)}$ are sparse and are non-zero only when $m_i = \pm m_o$, which further simplifies Eq. 4:

$$m_{ts}^{(L_o)} = \left(D^{(L_o)}\right)^{-1} \sum_{L_i,L_f} w_{L_i,L_f,L_o} \bigoplus_{m_o} \left(\left(\tilde{x}_s^{(L_i)}\right)_{m_o} C_{(L_i,m_o),(L_f,0)}^{(L_o,m_o)} + \left(\tilde{x}_s^{(L_i)}\right)_{-m_o} C_{(L_i,-m_o),(L_f,0)}^{(L_o,m_o)}\right)$$

$$(5)$$

By re-ordering the summations and concatenation in Eq. 5, we have:

$$\left(D^{(L_o)}\right)^{-1} \sum_{L_i} \bigoplus_{m_o} \left(\left(\tilde{x}_s^{(L_i)}\right)_{m_o} \sum_{L_f}\left(w_{L_i,L_f,L_o}C_{(L_i,m_o),(L_f,0)}^{(L_o,m_o)}\right) + \left(\tilde{x}_s^{(L_i)}\right)_{-m_o} \sum_{L_f}\left(w_{L_i,L_f,L_o}C_{(L_i,-m_o),(L_f,0)}^{(L_o,m_o)}\right)\right)$$

$$(6)$$

Instead of using learnable parameters for $w_{L_i,L_f,L_o}$, eSCN proposes to parametrize $\tilde{w}_{m_o}^{(L_i,L_o)}$ and $\tilde{w}_{-m_o}^{(L_i,L_o)}$ as below:

$$\tilde{w}_{m_o}^{(L_i,L_o)} = \sum_{L_f} w_{L_i,L_f,L_o}C_{(L_i,m_o),(L_f,0)}^{(L_o,m_o)} = \sum_{L_f} w_{L_i,L_f,L_o}C_{(L_i,-m_o),(L_f,0)}^{(L_o,-m_o)} \quad \text{for } m >= 0$$

$$\tilde{w}_{-m_o}^{(L_i,L_o)} = \sum_{L_f} w_{L_i,L_f,L_o}C_{(L_i,m_o),(L_f,0)}^{(L_o,-m_o)} = -\sum_{L_f} w_{L_i,L_f,L_o}C_{(L_i,-m_o),(L_f,0)}^{(L_o,m_o)} \quad \text{for } m > 0$$

$$(7)$$

The parametrization of $\tilde{w}_{m_o}^{(L_i,L_o)}$ and $\tilde{w}_{-m_o}^{(L_i,L_o)}$ enables removing the summation over $L_f$ and further simplifies the computation. By combining Eq. 6 and Eq. 7, we have:

$$m_{ts}^{(L_o)} = \left(D^{(L_o)}\right)^{-1} \sum_{L_i} \bigoplus_{m_o} \left(y_{ts}^{(L_i,L_o)}\right)_{m_o}$$

$$\left(y_{ts}^{(L_i,L_o)}\right)_{m_o} = \tilde{w}_{m_o}^{(L_i,L_o)}\left(\tilde{x}_s^{(L_i)}\right)_{m_o} - \tilde{w}_{-m_o}^{(L_i,L_o)}\left(\tilde{x}_s^{(L_i)}\right)_{-m_o} \quad \text{for } m_o > 0$$

$$\left(y_{ts}^{(L_i,L_o)}\right)_{-m_o} = \tilde{w}_{-m_o}^{(L_i,L_o)}\left(\tilde{x}_s^{(L_i)}\right)_{m_o} + \tilde{w}_{m_o}^{(L_i,L_o)}\left(\tilde{x}_s^{(L_i)}\right)_{-m_o} \quad \text{for } m_o > 0$$

$$\left(y_{ts}^{(L_i,L_o)}\right)_{m_o} = \tilde{w}_{m_o}^{(L_i,L_o)}\left(\tilde{x}_s^{(L_i)}\right)_{m_o} \quad \text{for } m_o = 0$$

$$(8)$$

The formulation of $y_{ts}^{(L_i, L_o)}$ coincides with performing $SO(2)$ linear operations (Worrall et al., 2016; Passaro & Zitnick, 2023). Additionally, eSCN convolutions can further simplify the computation by considering only a subset of $m_o$ components in Eq. 8, i.e., $|m_o| \leqslant M_{max}$.

In summary, efficient $SO(3)$ convolutions can be achieved by first rotating irreps features $x_s^{(L_i)}$ based on relative position vectors $\vec{r}_{ts}$ and then performing $SO(2)$ linear operations on rotated features. The key idea is that rotation simplifies the computation as in Eq. 4, 5, 7, and 8. Please refer to their work (Passaro & Zitnick, 2023) for more details. We note that eSCN convolutions consider only simplifying the case of taking tensor products between input irreps features and spherical harmonic projections of relative position vectors. eSCN convolutions do not simplify general cases such as taking tensor products between input irreps features and themselves (Batatia et al., 2022) since the relative position vectors used to rotate irreps features are not clearly defined.

## A.4 $S^2$ ACTIVATION

$S^2$ activation was first proposed in Spherical CNNs (Cohen et al., 2018). Our implementation of $S^2$ activation is the same as that in e3nn (Geiger et al., 2022), SCN (Zitnick et al., 2022) and eSCN (Passaro & Zitnick, 2023). Basically, we uniformly sample a fixed set of points on a unit sphere along the dimensions of longitude, parametrized by $\alpha \in [0, 2\pi)$, and latitude, parametrized by $\beta \in [0, \pi)$. We set the resolutions $R$ of $\alpha$ and $\beta$ to be 18 when $L_{max} = 6$, meaning that we will have 324 ($= 18 \times 18$) points. Once the points are sampled, they are kept the same during training and inference, and therefore there is no randomness. For each point on the unit sphere, we compute the spherical harmonics projection of degrees up to $L_{max}$. We consider an equivariant feature of $C$ channels and each channel contains vectors of all degrees from 0 to $L_{max}$. When performing $S^2$ activation, for each channel and for each sampled point, we first compute the inner product between the vectors of all degrees contained in one channel of the equivariant feature and the spherical harmonics projections of a sampled point. This results in $R \times R \times C$ values, where the first two dimensions, $R \times R$, correspond to grid resolutions and the last dimension corresponds to channels. They can be viewed as 2D grid feature maps and treated as scalars, and we can apply any standard or typical activation functions like SiLU or use standard linear layers performing feature aggregation along the channel dimension. After applying these functions, we project back to vectors of all degrees by multiplying those values with their corresponding spherical harmonics projections of sampled points. The process is the same as performing a Fourier transform, applying some functions and then performing an inverse Fourier transform.

Moreover, since the inner products between one channel of vectors of all degrees and the spherical harmonics projections of sampled points sum over all degrees, the conversion to 2D grid feature maps implicitly considers the information of all degrees. Therefore, $S^2$ activation, which converts equivariant features into 2D grid feature maps, uses the information of all degrees to determine the non-linearity. In contrast, gate activation only uses vectors of degree 0 to determine the non-linearity of vectors of higher degrees. For tasks such as force predictions, where the information of degrees is critical, $S^2$ activation can be better than gate activation since $S^2$ activation uses all degrees to determine non-linearity.

Although there is sampling on a sphere, the works (Cohen et al., 2018; Passaro & Zitnick, 2023) mention that as long as the number of samples, or resolution $R$, is high enough, the equivariance error can be close to zero. Furthermore, eSCN (Passaro & Zitnick, 2023) empirically computes such errors in Figure 9 in their latest manuscript and shows that the errors of using $L_{max} = 6$ and $R = 18$ are close to $0.2\%$, which is similar to the equivariance errors of tensor products in e3nn (Geiger et al., 2022). We note that the equivariance errors in e3nn are due to numerical precision.

## B RELATED WORKS

### B.1 *SE(3)/E(3)*-EQUIVARIANT GNNS

Equivariant neural networks (Thomas et al., 2018; Kondor et al., 2018; Weiler et al., 2018; Fuchs et al., 2020; Miller et al., 2020; Townshend et al., 2020; Batzner et al., 2022; Jing et al., 2021; Schütt et al., 2021; Satorras et al., 2021; Unke et al., 2021; Brandstetter et al., 2022; Thölke & Fabritiis, 2022; Le et al., 2022; Musaelian et al., 2022; Batatia et al., 2022; Liao & Smidt, 2023; Passaro & Zitnick, 2023)

use equivariant irreps features built from vector spaces of irreducible representations (irreps) to achieve equivariance to 3D rotation (Thomas et al., 2018; Weiler et al., 2018; Kondor et al., 2018). They operate on irreps features with equivariant operations like tensor products. Previous works differ in equivariant operations used in their networks and how they combine those operations. TFN (Thomas et al., 2018) and NequIP (Batzner et al., 2022) use equivariant graph convolution with linear messages built from tensor products, with the latter utilizing extra equivariant gate activation (Weiler et al., 2018). SEGNN (Brandstetter et al., 2022) introduces non-linearity to messages passing (Gilmer et al., 2017; Sanchez-Gonzalez et al., 2020) with equivariant gate activation, and the non-linear messages improve upon linear messages. SE(3)-Transformer (Fuchs et al., 2020) adopts equivariant dot product attention (Vaswani et al., 2017) with linear messages. Equiformer (Liao & Smidt, 2023) improves upon previously mentioned equivariant GNNs by combining MLP attention and non-linear messages. Equiformer additionally introduces equivariant layer normalization and regularizations like dropout (Srivastava et al., 2014) and stochastic depth (Huang et al., 2016). However, the networks mentioned above rely on compute-intensive $SO(3)$ tensor products to mix the information of vectors of different degrees during message passing, and therefore they are limited to small values for maximum degrees $L_{max}$ of equivariant representations. SCN (Zitnick et al., 2022) proposes rotating irreps features based on relative position vectors and identifies a subset of spherical harmonics coefficients, on which they can apply unconstrained functions. They further propose relaxing the requirement for strict equivariance and apply typical functions to rotated features during message passing, which trades strict equivariance for computational efficiency and enables using higher values of $L_{max}$. eSCN (Passaro & Zitnick, 2023) further improves upon SCN by replacing typical functions with $SO(2)$ linear layers for rotated features and imposing strict equivariance during message passing. However, except using more efficient operations for higher $L_{max}$, SCN and eSCN mainly adopt the same network design as SEGNN, which is less performant than Equiformer. In this work, we propose EquiformerV2, which includes all the benefits of the above networks by incorporating eSCN convolutions into Equiformer and adopts three additional architectural improvements.

## B.2 INVARIANT GNNs

Prior works (Schütt et al., 2017; Xie & Grossman, 2018; Unke & Meuwly, 2019; Gasteiger et al., 2020b;a; Qiao et al., 2020; Liu et al., 2022; Shuaibi et al., 2021; Klicpera et al., 2021; Sriram et al., 2022; Gasteiger et al., 2022) extract invariant information from 3D atomistic graphs and operate on the resulting graphs augmented with invariant features. Their differences lie in leveraging different geometric features such as distances, bond angles (3 atom features) or dihedral angles (4 atom features). SchNet (Schütt et al., 2017) models interaction between atoms with only relative distances. DimeNet series (Gasteiger et al., 2020b;a) use triplet representations of atoms to incorporate bond angles. SphereNet (Liu et al., 2022) and GemNet (Klicpera et al., 2021; Gasteiger et al., 2022) further include dihedral angles by considering quadruplet representations. However, the memory complexity of triplet and quadruplet representations of atoms do not scale well with the number of atoms, and this requires additional modifications like interaction hierarchy used by GemNet-OC (Gasteiger et al., 2022) for large datasets like OC20 (Chanussot* et al., 2021). Additionally, for the task of predicting DFT calculations of energies and forces on the large-scale OC20 dataset, invariant GNNs have been surpassed by equivariant GNNs recently.

## C DETAILS OF ARCHITECTURE

In this section, we define architectural hyper-parameters like maximum degrees and numbers of channels in certain layers in EquiformerV2, which are used to specify the detailed architectures in Sec. D.2, Sec. D.5, Sec. E.1 and Sec. F.2. Besides, we note that eSCN (Passaro & Zitnick, 2023) and this work mainly consider $SE(3)$-equivariance.

We denote embedding dimensions as $d_{embed}$, which defines the dimensions of most irreps features. Specifically, the output irreps features of all modules except the output head in Figure 1a have dimension $d_{embed}$. For separable $S^2$ activation as illustrated in Figure 2c, we denote the resolution of point samples on a sphere as $R$, which can depend on maximum degree $L_{max}$, and denote the unconstrained functions after projecting to point samples as $F$.

For equivariant graph attention in Figure 1b, the input irreps features $x_i$ and $x_j$ have dimension $d_{embed}$. The dimension of the irreps feature $f_{ij}^{(L)}$ is denoted as $d_{attn\_hidden}$. Equivariant graph

attention can have $h$ parallel attention functions. For each attention function, we denote the dimension of the scalar feature $f_{ij}^{(0)}$ as $d_{attn\_alpha}$ and denote the dimension of the value vector, which is in the form of irreps features, as $d_{attn\_value}$. For the separable $S^2$ activation used in equivariant graph attention, the resolution of point samples is $R$, and we use a single SiLU activation for $F$. We share the layer normalization in attention re-normalization across all $h$ attention functions but have different $h$ linear layers after that. The last linear layer projects the dimension back to $d_{embed}$. The two intermediate $SO(2)$ linear layers operate with maximum degree $L_{max}$ and maximum order $M_{max}$.

For feed forward networks (FFNs) in Figure 1d, we denote the dimension of the output irreps features of the first linear layer as $d_{ffn}$. For the separable $S^2$ activation used in FFNs, the resolution of point samples is $R$, and $F$ consists of a two-layer MLP, with each linear layer followed by SiLU, and a final linear layer. The linear layers have the same number of channels as $d_{ffn}$.

For radial functions, we denote the dimension of hidden scalar features as $d_{edge}$. For experiments on OC20, same as eSCN (Passaro & Zitnick, 2023), we use Gaussian radial basis to represent relative distances and additionally embed the atomic numbers at source nodes and target nodes with two scalar features of dimension $d_{edge}$. The radial basis and the two embeddings of atomic numbers are fed to the radial function to generate edge distance embeddings.

The maximum degree of irreps features is denoted as $L_{max}$. All irreps features have degrees from $0$ to $L_{max}$ and have $C$ channels for each degree. We denote the dimension as $(L_{max}, C)$. For example, irreps feature $x_{irreps}$ of dimension $(6, 128)$ has maximum degree $6$ and $128$ channels for each degree. The dimension of scalar feature $x_{scalar}$ can be expressed as $(0, C_{scalar})$.

Following Equiformer (Liao & Smidt, 2023), we apply dropout (Srivastava et al., 2014) to attention weights and stochastic depth (Huang et al., 2016) to outputs of equivariant graph attention and feed forward networks. However, we do not apply dropout or stochastic depth to the output head.

# D  DETAILS OF EXPERIMENTS ON OC20

## D.1  DETAILED DESCRIPTION OF OC20 DATASET

The large and diverse OC20 dataset (Chanussot* et al., 2021) (Creative Commons Attribution 4.0 License) consists of 1.2M DFT relaxations for training and evaluation, computed with the revised Perdew-Burke-Ernzerhof (RPBE) functional (Hammer et al., 1999). Each structure in OC20 has an adsorbate molecule placed on a catalyst surface, and the core task is Structure to Energy Forces (S2EF), which is to predict the energy of the structure and per-atom forces. Models trained for the S2EF task are evaluated on energy and force mean absolute error (MAE). These models can in turn be used for performing structure relaxations by using the model's force predictions to iteratively update the atomic positions until a relaxed structure corresponding to a local energy minimum is found. These relaxed structure and energy predictions are evaluated on the Initial Structure to Relaxed Structure (IS2RS) and Initial Structure to Relaxed Energy (IS2RE) tasks. The "All" split of OC20 contains 134M training structures spanning 56 elements, the "MD" split consists of 38M structures, and the "2M" split has 2M structures. For validation and test splits, there are four sub-splits containing in-distribution adsorbates and catalysts (ID), out-of-distribution adsorbates (OOD Ads), out-of-distribution catalysts (OOD Cat), and out-of-distribution adsorbates and catalysts (OOD Both).

## D.2  TRAINING DETAILS

**Hyper-Parameters.**  We summarize the hyper-parameters for the base model setting on OC20 S2EF-2M dataset and the main results on OC20 S2EF-All and S2EF-All+MD datasets in Table 7. For the ablation studies on OC20 S2EF-2M dataset, when trained for 20 or 30 epochs as in Table 1b, we increase the learning rate from $2 \times 10^{-4}$ to $4 \times 10^{-4}$. When using $L_{max} = 8$ as in Table 1c, we increase the resolution of point samples $R$ from 18 to 20. We vary $L_{max}$ and the widths for speed-accuracy trade-offs in Figure 4. Specifically, we first decrease $L_{max}$ from 6 to 4. Then, we multiply $h$ and the number of channels of $(d_{embed}, d_{attn\_hidden}, d_{ffn})$ by 0.75 and 0.5. We train all models for 30 epochs. The same strategy to scale down eSCN models is adopted for fair comparisons.

| Hyper-parameters | Base model setting on S2EF-2M | EquiformerV2 (31M) on S2EF-All+MD | EquiformerV2 (153M) on S2EF-All/S2EF-All+MD |
|---|---|---|---|
| Optimizer | AdamW | AdamW | AdamW |
| Learning rate scheduling | Cosine learning rate with linear warmup | Cosine learning rate with linear warmup | Cosine learning rate with linear warmup |
| Warmup epochs | 0.1 | 0.01 | 0.01 |
| Maximum learning rate | $2 \times 10^{-4}$ | $4 \times 10^{-4}$ | $4 \times 10^{-4}$ |
| Batch size | 64 | 512 | 256 for S2EF-All, 512 for S2EF-All+MD |
| Number of epochs | 12 | 3 | 1 |
| Weight decay | $1 \times 10^{-3}$ | $1 \times 10^{-3}$ | $1 \times 10^{-3}$ |
| Dropout rate | 0.1 | 0.1 | 0.1 |
| Stochastic depth | 0.05 | 0.1 | 0.1 |
| Energy coefficient $\lambda_E$ | 2 | 4 | 2 for S2EF-All, 2, 4 for S2EF-All+MD |
| Force coefficient $\lambda_F$ | 100 | 100 | 100 |
| Gradient clipping norm threshold | 100 | 100 | 100 |
| Model EMA decay | 0.999 | 0.999 | 0.999 |
| Cutoff radius (Å) | 12 | 12 | 12 |
| Maximum number of neighbors | 20 | 20 | 20 |
| Number of radial bases | 600 | 600 | 600 |
| Dimension of hidden scalar features in radial functions $d_{edge}$ | (0, 128) | (0, 128) | (0, 128) |
| Maximum degree $L_{max}$ | 6 | 4 | 6 |
| Maximum order $M_{max}$ | 2 | 2 | 3 |
| Number of Transformer blocks | 12 | 8 | 20 |
| Embedding dimension $d_{embed}$ | (6, 128) | (4, 128) | (6, 128) |
| $f_{ij}^{(L)}$ dimension $d_{attn\_hidden}$ | (6, 64) | (4, 64) | (6, 64) |
| Number of attention heads $h$ | 8 | 8 | 8 |
| $f_{ij}^{(0)}$ dimension $d_{attn\_alpha}$ | (0, 64) | (0, 64) | (0, 64) |
| Value dimension $d_{attn\_value}$ | (6, 16) | (4, 16) | (6, 16) |
| Hidden dimension in feed forward networks $d_{ffn}$ | (6, 128) | (4, 128) | (6, 128) |
| Resolution of point samples $R$ | 18 | 18 | 18 |

Table 7: Hyper-parameters for the base model setting on OC20 S2EF-2M dataset and the main results on OC20 S2EF-All and S2EF-All+MD datasets.

| Training set | Attention re-normalization | Activation | Normalization | $L_{max}$ | $M_{max}$ | Number of Transformer blocks | Training time (GPU-hours) | Inference speed (Samples / GPU sec.) | Number of parameters |
|---|---|---|---|---|---|---|---|---|---|
| | ✗ | Gate | LN | 6 | 2 | 12 | 965 | 19.06 | 91.06M |
| | ✓ | Gate | LN | 6 | 2 | 12 | 998 | 19.07 | 91.06M |
| | ✓ | $S^2$ | LN | 6 | 2 | 12 | 1476 | 12.80 | 81.46M |
| | ✓ | Sep. $S^2$ | LN | 6 | 2 | 12 | 1505 | 12.51 | 83.16M |
| | ✓ | Sep. $S^2$ | SLN | 6 | 2 | 12 | 1412 | 13.22 | 83.16M |
| S2EF-2M | ✓ | Sep. $S^2$ | SLN | 4 | 2 | 12 | 965 | 19.86 | 44.83M |
| | ✓ | Sep. $S^2$ | SLN | 8 | 2 | 12 | 2709 | 7.86 | 134.28M |
| | ✓ | Sep. $S^2$ | SLN | 6 | 3 | 12 | 1623 | 11.92 | 95.11M |
| | ✓ | Sep. $S^2$ | SLN | 6 | 4 | 12 | 2706 | 7.98 | 102.14M |
| | ✓ | Sep. $S^2$ | SLN | 6 | 6 | 12 | 3052 | 7.13 | 106.63M |
| S2EF-All | ✓ | Sep. $S^2$ | SLN | 6 | 3 | 20 | 20499 | 6.08 | 153.60M |
| S2EF-All+MD ($\lambda_E = 2$) | ✓ | Sep. $S^2$ | SLN | 6 | 3 | 20 | 32834 | 6.08 | 153.60M |
| S2EF-All+MD ($\lambda_E = 4$) | ✓ | Sep. $S^2$ | SLN | 4 | 2 | 8 | 16931 | 29.21 | 31.06M |
| S2EF-All+MD ($\lambda_E = 4$) | ✓ | Sep. $S^2$ | SLN | 6 | 3 | 20 | 37692 | 6.08 | 153.60M |

Table 8: Training time, inference speed and numbers of parameters of different models trained on OC20 S2EF-2M, S2EF-All and S2EF-All+MD datasets. All numbers are measured on V100 GPUs with 32GB.

**Training Time, Inference Speed and Numbers of Parameters.** Table 8 summarizes the training time, inference speed and numbers of parameters of models in Tables 1a (Index 1, 2, 3, 4, 5), 1c, 1d and 2. V100 GPUs with 32GB are used to train all models. We use 16 GPUs to train each individual model on S2EF-2M dataset, 64 GPUs for S2EF-All, 64 GPUs for EquiformerV2 (31M) on S2EF-All+MD, and 128 GPUs for EquiformerV2 (153M) on S2EF-All+MD.

## D.3 DETAILS OF RUNNING RELAXATIONS

A structural relaxation is a local optimization where atom positions are iteratively updated based on forces to minimize the energy of the structure. We perform ML relaxations using the LBFGS optimizer (quasi-Newton) implemented in the Open Catalyst Github repository (Chanussot* et al., 2021). The structural relaxations for OC20 IS2RE and IS2RS tasks are allowed to run for 200 steps or until the maximum predicted force per atom $F_{max} \leqslant 0.02$ eV/Å , and the relaxations for AdsorbML are allowed to run for 300 steps or until $F_{max} \leqslant 0.02$ eV/Å . These settings are chosen to be consistent with prior works. We run relaxations on V100 GPUs with 32GB. The computational cost of running relaxations with EquiformerV2 (153M) for OC20 IS2RE and IS2RS tasks is 1011 GPU-hours, and that of running ML relaxations for AdsorbML is 1075 GPU-hours. The time for

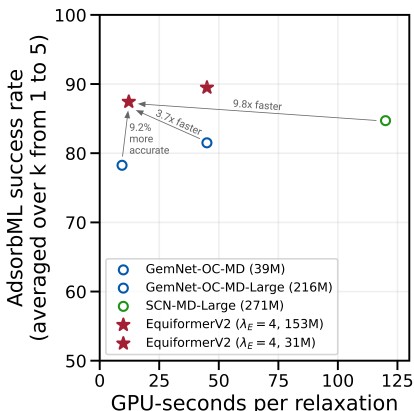

Figure 5: Speed-accuracy trade-offs of different models when used in the AdsorbML algorithm.

running relaxations with EquiformerV2 (31M) is 240 GPU-hours for OC20 IS2RE and IS2RS and 298 GPU-hours for AdsorbML.

### D.4 DETAILS OF ADSORBML

We run the AdsorbML algorithm on the OC20-Dense dataset in accordance with the procedure laid out in the paper (Lan et al., 2022), which is summarized here:

1. Run ML relaxations on all initial structures in the OC20-Dense dataset. There are around 1000 different adsorbate-surface combinations with about 90 adsorbate placements per combination, and therefore we have roughly 90k structures in total.

2. Remove invalid ML relaxed structures based on physical constraints and rank the other ML relaxed structures in order of lowest to highest ML predicted energy.

3. Take the top $k$ ML relaxed structures with the lowest ML predicted energies for each adsorbate-surface combination and run DFT single-point calculations. The single-point calculations are performed on the ML relaxed structures to improve the energy predictions without running a full DFT relaxation and are run with VASP using the same setting as the original AdsorbML experiments. As shown in Table 3, we vary $k$ from 1 to 5.

4. Compute success and speedup metrics based on our lowest DFT single-point energy per adsorbate-surface combination and the DFT labels provided in the OC20-Dense dataset.

To better understand the speed-accuracy trade-offs of different models, we compare the AdsorbML success rate averaged over $k$ from 1 to 5 and average GPU-seconds of running one structure relaxation in Figure 5. We visualize some examples of relaxed structures from eSCN (Passaro & Zitnick, 2023), EquiformerV2 and DFT in Figure 6.

### D.5 ADDITIONAL COMPARISON WITH EQUIFORMER ON OC20 IS2RE

**Training Details.** We follow the same setting as Equiformer (Liao & Smidt, 2023) and train two EquiformerV2 models on OC20 IS2RE dataset without and with IS2RS auxiliary task. We use the same radial basis function as Equiformer. When IS2RS auxiliary task is adopted, we use a linearly decayed weight for loss associated with IS2RS, which starts at 15 and decays to 1 and adopt Noisy Nodes data augmentation (Godwin et al., 2022). The hyper-parameters are summarized in Table 9. We train EquiformerV2 on OC20 IS2RE with 16 V100 GPUs with 32GB and train on OC20 IS2RE with IS2RS auxiliary task and Noisy Nodes data augmentation with 32 V100 GPUs. The training costs are 574 and 2075 GPU-hours, and the numbers of parameters are 36.03M and 95.24M.

**Results.** The comparison is shown in Table 10. Without IS2RS auxiliary task, EquiformerV2 overfits the training set due to higher degrees and achieves worse results than Equiformer. However, with IS2RS auxiliary task and Noisy Nodes data augmentation, EquiformerV2 achieves better energy MAE. The different rankings of models under different settings is also found in Noisy Nodes (Godwin

| | EquiformerV2 on OC20 IS2RE | EquiformerV2 on OC20 IS2RE with IS2RS auxiliary task and Noisy Nodes |
|---|---|---|
| Optimizer | AdamW | AdamW |
| Learning rate scheduling | Cosine learning rate with linear warmup | Cosine learning rate with linear warmup |
| Warmup epochs | 2 | 2 |
| Maximum learning rate | $4 \times 10^{-4}$ | $5 \times 10^{-4}$ |
| Batch size | 32 | 64 |
| Number of epochs | 20 | 40 |
| Weight decay | $1 \times 10^{-3}$ | $1 \times 10^{-3}$ |
| Dropout rate | 0.2 | 0.2 |
| Stochastic depth | 0.05 | 0.1 |
| Gradient clipping norm threshold | 100 | 100 |
| Model EMA decay | 0.999 | 0.999 |
| Cutoff radius (Å) | 5.0 | 5.0 |
| Maximum number of neighbors | 50 | 50 |
| Number of radial basis | 128 | 128 |
| Dimension of hidden scalar features in radial functions $d_{edge}$ | $(0, 64)$ | $(0, 64)$ |
| Maximum degree $L_{max}$ | 6 | 6 |
| Maximum order $M_{max}$ | 2 | 2 |
| Number of Transformer blocks | 6 | 15 |
| Embedding dimension $d_{embed}$ | $(6, 128)$ | $(6, 128)$ |
| $f_{ij}^{(L)}$ dimension $d_{attn\_hidden}$ | $(6, 64)$ | $(6, 64)$ |
| Number of attention heads $h$ | 8 | 8 |
| $f_{ij}^{(0)}$ dimension $d_{attn\_alpha}$ | $(0, 64)$ | $(0, 64)$ |
| Value dimension $d_{attn\_value}$ | $(6, 16)$ | $(6, 16)$ |
| Hidden dimension in feed forward networks $d_{ffn}$ | $(6, 128)$ | $(6, 128)$ |
| Resolution of point samples $R$ | 18 | 18 |

Table 9: Hyper-parameters for OC20 IS2RE dataset.

| | IS2RE direct | | | | | IS2RE direct with IS2RS auxiliary task and Noisy Nodes | | | | |
|---|---|---|---|---|---|---|---|---|---|---|
| | Energy MAE (meV)↓ | | | | | Energy MAE (meV)↓ | | | | |
| Model | ID | OOD Ads | OOD Cat | OOD Both | Average | ID | OOD Ads | OOD Cat | OOD Both | Average |
| Equiformer (Liao & Smidt, 2023) | **508.8** | **627.1** | **505.1** | **554.5** | **548.9** | 415.6 | 497.6 | 416.5 | 434.4 | 441.0 |
| EquiformerV2 | 516.1 | 704.1 | 524.5 | 636.5 | 595.3 | **400.4** | **459.0** | **406.2** | **401.8** | **416.9** |

Table 10: OC20 IS2RE results on the validation split. With IS2RS auxiliary task and Noisy Nodes (Godwin et al., 2022) data augmentation, EquiformerV2 achieves better energy MAE.

et al., 2022), where they mention a node-level auxiliary task can prevent overfitting and enable more expressive models to perform better.

# E   DETAILS OF EXPERIMENTS ON OC22

## E.1   TRAINING DETAILS

The hyper-parameters for OC22 dataset is summarized in Table 11. We use 32 V100 GPUs with 32GB and train two EquiformerV2 models with different energy coefficients $\lambda_E$ and force coefficeints $\lambda_F$. The number of parameters is 121.53M, and the training cost of each model is 4552 GPU-hours. The time for running relaxations for OC22 IS2RE is 38 GPU-hours.

# F   DETAILS OF EXPERIMENTS ON QM9

## F.1   ADDITIONAL RESULTS OF TRAINING WITH NOISY NODES

Similar to Sec. D.5, we train EquiformV2 on QM9 with Noisy Nodes (Godwin et al., 2022) to show that the performance gain brought by using higher degrees can be larger when trained with a node-level auxiliary task and data augmentation. In Table 12, we summarize the results and compare with previous works using Noisy Nodes and pre-training via denoising (Zaidi et al., 2023). When trained with Noisy Nodes, EquiformerV2 performs better than Equiformer on more tasks. Specifically, without Noisy Nodes, EquiformerV2 is better than Equiformer on 9 out of the 12 tasks, similar on 1 task, and worse on the other 2 tasks. With Noisy Nodes, EquiformerV2 achieves better MAE on 10 tasks, similar on 1 task, and worse on 1 task. Additionally, we note that EquiformerV2 with Noisy Nodes is overall better than GNS-TAT with both pre-training and Noisy Nodes (Zaidi et al., 2023) on 9 out of the 12 tasks even though EquiformerV2 is not pre-trained on PCQM4Mv2 dataset,

| Hyper-parameters | Value or description |
|---|---|
| Optimizer | AdamW |
| Learning rate scheduling | Cosine learning rate with linear warmup |
| Warmup epochs | 0.1 |
| Maximum learning rate | $2 \times 10^{-4}$ |
| Batch size | 128 |
| Number of epochs | 6 |
| Weight decay | $1 \times 10^{-3}$ |
| Dropout rate | 0.1 |
| Stochastic depth | 0.1 |
| Energy coefficient $\lambda_E$ | 1, 4 |
| Force coefficient $\lambda_F$ | 1, 100 |
| Gradient clipping norm threshold | 50 |
| Model EMA decay | 0.999 |
| Cutoff radius (Å) | 12 |
| Maximum number of neighbors | 20 |
| Number of radial bases | 600 |
| Dimension of hidden scalar features in radial functions $d_{edge}$ | (0, 128) |
| Maximum degree $L_{max}$ | 6 |
| Maximum order $M_{max}$ | 2 |
| Number of Transformer blocks | 18 |
| Embedding dimension $d_{embed}$ | (6, 128) |
| $f_{ij}^{(L)}$ dimension $d_{attn\_hidden}$ | (6, 64) |
| Number of attention heads $h$ | 8 |
| $f_{ij}^{(0)}$ dimension $d_{attn\_alpha}$ | (0, 64) |
| Value dimension $d_{attn\_value}$ | (6, 16) |
| Hidden dimension in feed forward networks $d_{ffn}$ | (6, 128) |
| Resolution of point samples $R$ | 18 |

Table 11: Hyper-parameters for OC22 dataset.

| Model | Task Units | $\alpha$ $a_0^3$ | $\Delta\varepsilon$ meV | $\varepsilon_{HOMO}$ meV | $\varepsilon_{LUMO}$ meV | $\mu$ D | $C_\nu$ cal/mol K | $G$ meV | $H$ meV | $R^2$ $a_0^2$ | $U$ meV | $U_0$ meV | ZPVE meV |
|---|---|---|---|---|---|---|---|---|---|---|---|---|---|
| Equiformer (Liao & Smidt, 2023) | | .046 | 30 | 15 | 14 | .011 | .023 | 7.63 | 6.63 | .251 | 6.74 | 6.59 | 1.26 |
| Equiformer (Liao & Smidt, 2023) + NN[†] | | .040 | 26.4 | 13.7 | 13.0 | .011 | **.020** | 5.49 | 4.61 | .235 | 4.81 | 4.61 | 1.18 |
| GNS-TAT + NN (Zaidi et al., 2023)[‡] | | .047 | 25.7 | 17.3 | 17.1 | .021 | .022 | 7.41 | 6.42 | 0.65 | 6.39 | 6.39 | 1.080 |
| GNS-TAT + NN + pretraining (Zaidi et al., 2023)[‡] | | .040 | **22.0** | 14.9 | 14.7 | .016 | **.020** | 6.90 | 5.79 | 0.44 | 5.76 | 5.76 | **1.018** |
| EquiformerV2 | | .050 | 29 | 14 | 13 | .010 | .023 | 7.57 | 6.22 | .186 | 6.49 | 6.17 | 1.47 |
| EquiformerV2 + NN | | **.039** | 24.2 | **12.2** | **11.4** | **.009** | **.020** | **5.34** | **4.24** | **.182** | **4.28** | **4.34** | 1.21 |

Table 12: Mean absolute error results on QM9 test set when trained with Noisy Nodes (Godwin et al., 2022). † denotes that the results are produced by this work. ‡ denotes using different data partitions. "NN" denotes Noisy Nodes, and "pretraining" denotes pretraining on PCQM4Mv2 dataset (Nakata & Shimazaki, 2017). The performance gain from Equiformer to EquiformerV2 becomes larger when trained with Noisy Nodes.

which is more than $30\times$ larger than QM9. This shows that a more expressive model can match the performance with significantly less data.

## F.2  TRAINING DETAILS

We follow the data partition of Equiformer. For the tasks of $\mu$, $\alpha$, $\varepsilon_{HOMO}$, $\varepsilon_{LUMO}$, $\Delta\varepsilon$, and $C_\nu$, we use batch size $= 64$, the number of epochs $= 300$, learning rate $= 5 \times 10^{-4}$, Gaussian radial basis functions with the number of bases $= 128$, the number of Transformer blocks $= 6$, weight decay $= 5 \times 10^{-3}$, and dropout rate $= 0.2$ and use mixed precision for training. For the task of $R^2$, we use batch size $= 48$, the number of epochs $= 300$, learning rate $= 1.5 \times 10^{-4}$, Gaussian radial basis functions with the number of bases $= 128$, the number of Transformer blocks $= 5$, weight decay $= 5 \times 10^{-3}$, and dropout rate $= 0.1$ and use single precision for training. For the task of ZPVE, we use batch size $= 48$, the number of epochs $= 300$, learning rate $= 1.5 \times 10^{-4}$, Gaussian radial basis functions with the number of bases $= 128$, the number of Transformer blocks $= 5$, weight decay $= 5 \times 10^{-3}$, and dropout rate $= 0.2$ and use single precision for training. For the task of $G$, $H$, $U$, and $U_0$, we use batch size $= 48$, the number of epochs $= 300$, learning rate $= 1.5 \times 10^{-4}$, Gaussian radial basis functions with the number of bases $= 128$, the number of Transformer blocks $= 5$, weight decay $= 0.0$, and dropout rate $= 0.0$ and use single precision for training. Other hyper-parameters are the same across all the tasks, and we summarize them in Table 13. We use a single A6000 GPU and train different models for different tasks. The training costs are 72 GPU-hours for mixed precision training and 137 GPU-hours for single precision training. The number of parameters are 11.20M for 6 blocks and 9.35M for 5 blocks.

As for training with Noisy Nodes as mentioned in Sec. F.1, we add noise to atomic coordinates and incorporate a node-level auxiliary task of denoising atomic coordinates. We thus introduce

| Hyper-parameters | Value or description |
|---|---|
| Optimizer | AdamW |
| Learning rate scheduling | Cosine learning rate with linear warmup |
| Warmup epochs | 5 |
| Maximum learning rate | $1.5 \times 10^{-4}, 5 \times 10^{-4}$ |
| Batch size | $48, 64$ |
| Number of epochs | 300 |
| Weight decay | $0.0, 5 \times 10^{-3}$ |
| Dropout rate | $0.0, 0.1, 0.2$ |
| Stochastic depth | $0.0, 0.05$ |
| Cutoff radius (Å) | 5.0 |
| Maximum number of neighbors | 500 |
| Number of radial bases | 128 |
| Dimension of hidden scalar features in radial functions $d_{edge}$ | $(0, 64)$ |
| Maximum degree $L_{max}$ | 4 |
| Maximum order $M_{max}$ | 4 |
| Number of Transformer blocks | $5, 6$ |
| Embedding dimension $d_{embed}$ | $(4, 96)$ |
| $f_{ij}^{(L)}$ dimension $d_{attn\_hidden}$ | $(4, 48)$ |
| Number of attention heads $h$ | 4 |
| $f_{ij}^{(0)}$ dimension $d_{attn\_alpha}$ | $(0, 64)$ |
| Value dimension $d_{attn\_value}$ | $(4, 24)$ |
| Hidden dimension in feed forward networks $d_{ffn}$ | $(4, 96)$ |
| Resolution of point samples $R$ | 18 |
| Noise standard deviation $\sigma_{\text{denoise}}$ | 0.02 |
| Denoising coefficient $\lambda_{\text{denoise}}$ | 0.1 |
| Denoising probability $p_{\text{denoise}}$ | 0.5 |
| Corrupt ratio $r_{\text{denoise}}$ | $0.125, 0.25$ |

Table 13: Hyper-parameters for QM9 dataset.

four additional hyper-parameters, which are noise standard deviation $\sigma_{\text{denoise}}$, denoising coefficient $\lambda_{\text{denoise}}$, denoising probability $p_{\text{denoise}}$ and corrupt ratio $r_{\text{denoise}}$. The noise standard deviation $\sigma_{\text{denoise}}$ denotes the standard deviation of Gaussian noise added to each xyz component of atomic coordinates. The denoising coefficient $\lambda_{\text{denoise}}$ controls the relative importance of the auxiliary task compared to the original task. The denoising probability $p_{\text{denoise}}$ denotes the probability of adding noise to atomic coordinates and optimizing for both the auxiliary task and the original task. Using $p_{\text{denoise}} < 1$ enables taking original atomistic structures without any noise as inputs and optimizing for only the original task for some training iterations. The corrupt ratio $r_{\text{denoise}}$ denotes the ratio of the number of atoms, which we add noise to and denoise, to the total number of atoms. Using $r_{\text{denoise}} < 1$ allows only adding noise to and denoising a subset of atoms within a structure. For the task of $R^2$, we use $\sigma_{\text{denoise}} = 0.02$, $\lambda_{\text{denoise}} = 0.1$, $p_{\text{denoise}} = 0.5$ and $r_{\text{denoise}} = 0.125$. For other tasks, we use $\sigma_{\text{denoise}} = 0.02$, $\lambda_{\text{denoise}} = 0.1$, $p_{\text{denoise}} = 0.5$ and $r_{\text{denoise}} = 0.25$. We share the above hyper-parameters for training EquiformerV2 and Equiformer, and we add one additional block of equivariant graph attention for the auxiliary task. We slightly tune other hyper-parameters when trained with Noisy Nodes. For Equiformer, we additionally use stochastic depth = 0.05 for the tasks of $\alpha, \Delta\varepsilon, \varepsilon_{\text{HOMO}}, \varepsilon_{\text{LUMO}}$, and $C_\nu$. As for EquiformerV2, we additionally use stochastic depth = 0.05 for the tasks of $\mu, \Delta\varepsilon, \varepsilon_{\text{HOMO}}, \varepsilon_{\text{LUMO}}$, and $C_\nu$. We increase the number of blocks from 5 to 6 and increase the batch size from 48 to 64 for the tasks of $G, H, U$, and $U_0$. We increase the learning rate from $1.5 \times 10^{-4}$ to $5 \times 10^{-4}$ and increase the number of blocks from 5 to 6 for the task of $R^2$.

### F.3 ABLATION STUDY ON ARCHITECTURAL IMPROVEMENTS

We conduct ablation studies on the proposed architectural improvements using the task of $\Delta\varepsilon$ on QM9 and compare with Equiformer baseline (Liao & Smidt, 2023). The reults are summarized in Table 14. The comparison between Index 0 and Index 1 shows that directly increasing $L_{max}$ from 2 to 4 and using eSCN convolutions degrade the performance. This is due to overfitting since the QM9 dataset is smaller, and each structure in QM9 has fewer atoms, less diverse atom types and much less angular variations than OC20 and OC22. Comparing Index 1 and Index 2, attention re-normalization clearly improves the MAE result. Although using $S^2$ activation is stable here (Index 3) unlike OC20, it results in higher error than using gate activation (Index 2) and the Equiformer baseline (Index 0). When using the proposed separable $S^2$ activation (Index 4), we achieve lower error than using gate activation (Index 2). We can further reduce the error by using the proposed separable layer normalization (Index 5). Comparing Index 0 and Index 5, we note that the proposed architectural improvements are necessary to achieve better results than the baseline when using higher degrees on QM9. Overall, these ablation results follow the same trends as OC20.

| Index | Attention re-normalization | Activation | Normalization | $L_{max}$ | $\Delta\varepsilon$ MAE (meV) |
|---|---|---|---|---|---|
| 0 | Equiformer baseline | | | 2 | 29.98 |
| 1 | ✗ | Gate | LN | 4 | 30.46 |
| 2 | ✓ | Gate | LN | 4 | 29.51 |
| 3 | ✓ | $S^2$ | LN | 4 | 30.23 |
| 4 | ✓ | Sep. $S^2$ | LN | 4 | 29.31 |
| 5 | ✓ | Sep. $S^2$ | SLN | 4 | 29.03 |

Table 14: Ablation studies on the proposed architectural improvements using the task of $\Delta\varepsilon$ of the QM9 dataset.

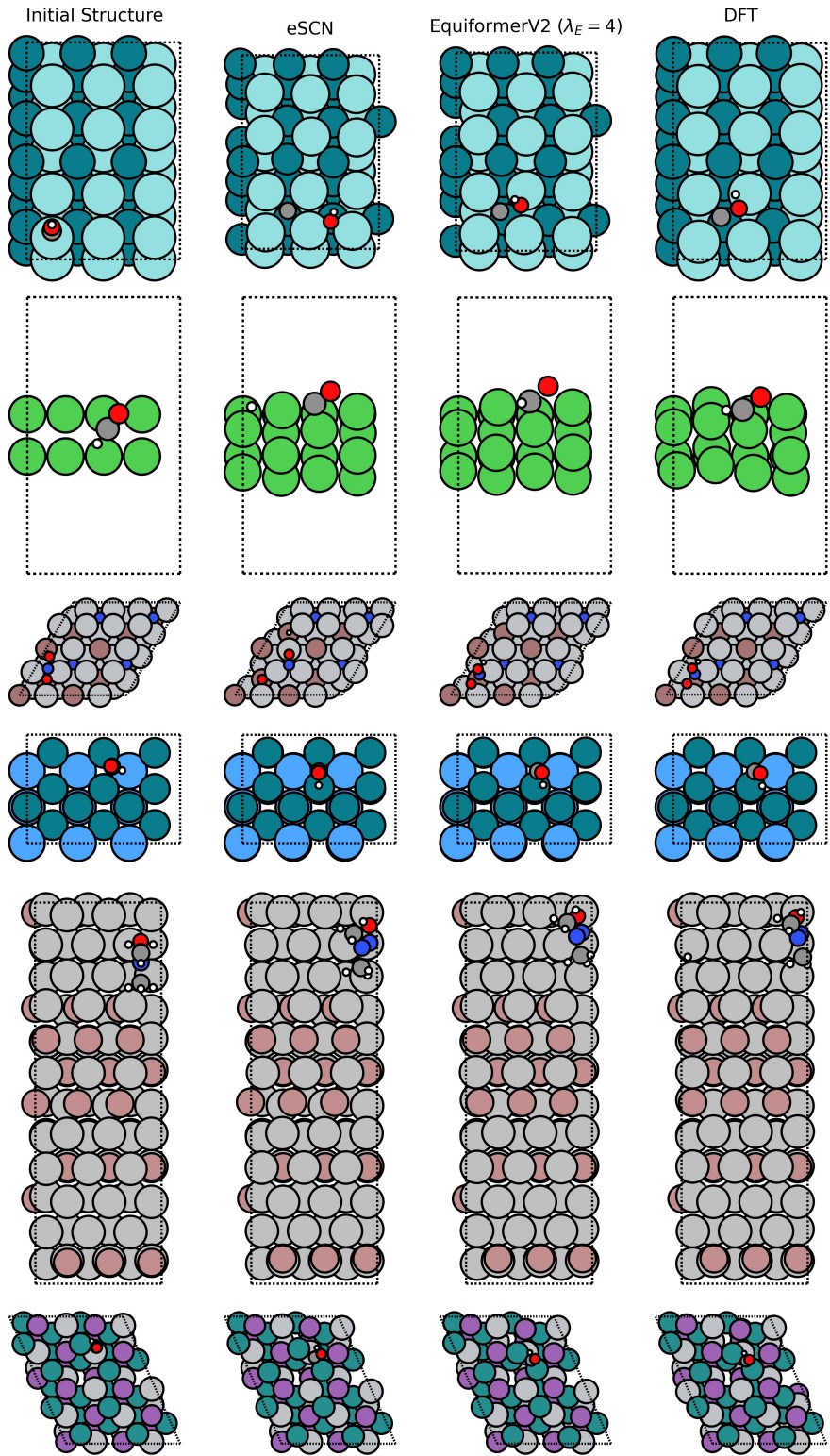

Figure 6: Qualitative examples of the initial configuration of an adsorbate on a catalyst surface (column 1), and corresponding relaxed configurations obtained from eSCN (Passaro & Zitnick, 2023) (column 2), EquiformerV2 (column 3), and DFT (column 4). All examples are selected from the OC20-Dense dataset (Lan et al., 2022). We show top-down views of each structure, with dashed lines showing the boundary of the unit cell repeating in the $x$ and $y$ directions.

