# OpenReview forum: "EquiformerV2: Improved Equivariant Transformer for Scaling to Higher-Degree Representations"
_ICLR.cc/2024/Conference — ICLR 2024 poster_

### Official Review · Reviewer_c4fs · 2023-10-31

**Soundness:** 3 good
**Presentation:** 2 fair
**Contribution:** 3 good
**Rating:** 6
**Confidence:** 5

**Summary:**

In this paper, the authors proposed EquiformerV2, which is a newly developed equivariant network for 3D molecular modeling built on the Equiformer. From the experimental evaluation, the EquiformerV2 model achieved strong performance on the large-scale OC20/OC22 benchmark, QM9 dataset and also the new AdsorbML dataset. Such performance improvement upon Equiformer is achieved via several architectural modifications: (1) levering eSCN's efficient SO(2) convolution implementation for SO(3) convolutions (tensor product operations); (2) Attention Re-normalization for stabilizing training; (3) Separable S2 activation for mixing representations with different degrees; (4) separate Layer Normalization.

**Strengths:**

1. **Regarding the problem studied in this paper**. By leveraging the key techniques from eSCN, the EquiformerV2 also achieves learning irreducible representations with larger maximum degrees, which has been verified again to be useful for large-scale DFT benchmarks.

2. **Regarding the empirical performance**. In the OC20 benchmark, EquiformerV2 sets a new standard by delivering state-of-the-art performance in the Structure-to-Energy-Force task. The model, further trained on this task, effectively serves as a force-field evaluator, demonstrating impressive performance in both IS2RS and IS2RE tasks. EquiformerV2 surpasses the performance of the compared baselines across all tasks, with a notable edge in force prediction. Furthermore, it significantly enhances the success rate on the AdsorbML dataset.

**Weaknesses:**

The novelty of the proposed architectural modifications is limited. Both the efficient SO(2) convolution and S^2 activation are from eSCN, while the attention re-normalization and layer normalization are more like engineering tricks. Among these differences from Equiformer, the eSCN SO(2) convolution plays an essential role in enabling the use of irreducible representations of higher degrees, and the S^2 activation also replaces all non-linear activations. In fact, these design strategies should be mainly credited to the eSCN work.

*******************************************Post Rebuttal *********************************************

Thank the authors for the response. I choose to keep my positive evaluation and hope the authors carefully include the newly added discussion and results in the final version of this paper.

**Questions:**

See the comments in the Weaknesses section

---

> ### Author Response · Authors · 2023-11-22
> **Response to Reviewer c4fs (1/1)**
>
> We thanks the reviewer for helpful feedback and address the comments below.
>
> ---
>
> > 1. [Weakness 1] The novelty of the proposed architectural modifications is limited.
>
> Please see **General Response 1**.
>
> ---
>
> > 2. [Weakness 2] Both the efficient SO(2) convolution and $S^2$ activation are from eSCN. … These design strategies should be mainly credited to the eSCN work.
>
> Indeed, we do credit and cite prior works in the paper — eSCN [1] for proposing SO(2) convolution, and Spherical CNNs [2] for $S^2$ activation (which was later adopted by SCN [3] and eSCN [1]).
> However, naively combining eSCN convolution with Equiformer does not work well (Table 1(a)), and naively using $S^2$ activation is unstable to train. Our proposed modification of **separable** $S^2$ activation is necessary since it stabilizes training on the OC20 S2EF-2M dataset (Index 3 and Index 4 in Table 1(a)).
>
> Reference:
> [1] Passaro et al. Reducing SO(3) Convolutions to SO(2) for Efficient Equivariant GNNs. ICML 2023.
> [2] Cohen et al. Spherical CNNs. ICLR 2018.
> [3] Zitnick et al. Spherical Channels for Modeling Atomic Interactions. NeurIPS 2022.
>
> ---
>
> > 3. [Weakness 3] The attention re-normalization and layer normalization are more like engineering tricks.
>
> The three proposed modifications are conceptually simple but necessary to extract the most performance gain when using higher degree representations. Naively scaling to higher degrees without these does not improve and can sometimes even hurt performance. We have shown this through results on OC20 (Table 1(a)) and QM9 (Table 13 in the revision).

---

### Official Review · Reviewer_3vjk · 2023-10-31

**Soundness:** 4 excellent
**Presentation:** 3 good
**Contribution:** 3 good
**Rating:** 6
**Confidence:** 4

**Summary:**

EquiformerV2 is proposed to improve the efficiency of Equiformer on higher-degree tensors.
To achieve this, original tensor product (TP) with spherical harmonics is changed to eSCN convolution which can reduce the complexity from $O(L^6)$ to $O(L^3)$.
Besides, three archtecture module is replaced to improve the performance in attention normalization, nonlinear activation and layer normalization.

**Strengths:**

The empirical results of EquiformerV2 is great. It achieves SOTA performance on OC20 and OC22, where higher-degree tensor shows great improvement. Meanwhile, the efficiency is denoted in Figure 4 showing that EquiformerV2 can has better efficient ability than eSCN.

**Weaknesses:**

The modification of proposed architecture is similar to the previous Equiformer. Although the ablation studies show the improvement of proposed modules, the results on QM9 is similar compared to Equiformer.

**Questions:**

Minor issue:
There is a double citation. Gao Huang, Yu Sun, Zhuang Liu, Daniel Sedra, and Kilian Q. Weinberger. Deep networks with stochastic depth. In European Conference on Computer Vision (ECCV), 2016a.

---

> ### Author Response · Authors · 2023-11-22
> **Response to Reviewer 3vjk (1/1)**
>
> We thank the reviewer for helpful feedback and address the comments below.
>
> ---
>
> > 1. [Weakness 1] The modification of proposed architecture is similar to the previous Equiformer.
>
> Please see **General Response 1**.
>
> ---
>
> > 2. [Weakness 2] The results on QM9 are similar compared to Equiformer.
>
> We disagree. Reiterating results from Table 5 in the paper, EquiformerV2 achieves better results than Equiformer on 9 out of 12 tasks on QM9, is similar on 1 task, and worse on 2.
>
> As for the much larger and diverse OC20 dataset, EquiformerV2 is considerably better (Table 6), by up to 25.8% on forces and 4.2% on energies.
>
> In Section 4.3, we do acknowledge that the performance gain from using higher degrees and the improved architecture are not as significant on QM9 as they are on OC20. This is consistent with prior work, which has shown that trends on small and large datasets are not always consistent [1]. Due to the small size of QM9, and limited diversity in atom types, counts and angular variation, higher degree networks are more prone to overfitting. In Section 4.3, we also outline potential ways of mitigating this overfitting by first pre-training on large datasets and then transferring to smaller datasets, as has recently been demonstrated by [2].
>
> Reference:
>
> [1] Gasteiger et al. GemNet-OC: Developing Graph Neural Networks for Large and Diverse Molecular Simulation Datasets. TMLR 2022.
>
> [2] Shoghi et al. From Molecules to Materials: Pre-training Large Generalizable Models for Atomic Property Prediction. ArXiv 2023.
>
> ---
>
> > 3. [Question 1] There is a double citation.
>
> Thanks! We correct this in the revision.

---

### Official Review · Reviewer_Joeu · 2023-11-07

**Soundness:** 3 good
**Presentation:** 3 good
**Contribution:** 2 fair
**Rating:** 6
**Confidence:** 4

**Summary:**

The authors propose EquiformerV2, which incorporates eSCN convolutions to efficiently include higher-degree tensors and introduces three architectural improvements: attention re-normalization, separable $S^2$ activation, and separable layer normalization. These enhancements allow EquiformerV2 to outperform state-of-the-art methods across OC20 and OC22.

**Strengths:**

- One of the significant contributions of this paper is the comprehensive experiments across OC20, OC22, and QM9. And EquiformerV2 achieves the state-of-the-art result over OC20 and OC22. The authors deserve commendation for their efforts in this aspect.
- The use of attention re-normalization, separable $S^2$ activation, and separable layer normalization is novel.

**Weaknesses:**

Major:
- Although the authors did a fantastic job on the experiments, EquiformerV2 is an incremental improvement over existing methods of both eSCN and Equiformer w.r.t. theory. And the novelty lies in those three specific techniques and enhancements. To see if these techniques are generalizable, I would like to see the ablation study of attention re-normalization, separable $S^2$ activation, and separable layer normalization, respectively, on the QM9 dataset like what the authors did in Table (a) for OC20.

Minors:
- Equation (2) in Appendix A.1: Use $\ddots$ instead of $\dots$
- Equation (4) in Appendix A.3: Commonly, the left side of an equation is used for assigning new notation. I recommend write $D^{(L)} = D^{(L)}(R_{ts})$ and $\tilde{x}_s^{(L)} = D^{(L)} x_s^{(L)}$ for a degree $L$ before Equation (4).

**Questions:**

See weaknesses.

---

> ### Author Response · Authors · 2023-11-22
> **Response to Reviewer Joeu (1/1)**
>
> We thank the reviewer for helpful feedback and address the comments below.
>
> ---
>
> > 1. [Major Weakness 1] Ablation study of attention re-normalization, separable $S^2$ activation, and separable layer normalization, on the QM9 dataset.
>
> We conduct ablation studies similar to Table 1(a) using the task of $\Delta \varepsilon$ on QM9 and compare with the Equiformer baseline [1]. The mean absolute error (MAE) results are as below.
>
> | Index  | Attention re-normalization |   Activation  |     Normalization    | $L_{max}$ | MAE (meV) |
> |:------:|:--------------------------:|:-------------:|:--------------------:|:-------:|:---------:|
> | 0      | Equiformer baseline        |               |                      | 2       | 29.98     |
> | 1      |                            |      Gate     |      Layer norm      |    4    |   30.46   |
> | 2      |              ✔             |      Gate     |      Layer norm      |    4    |   29.51   |
> | 3      |              ✔             |      $S^2$      |      Layer norm      |    4    |   30.23   |
> | 4      |              ✔             | Separable $S^2$ |      Layer norm      |    4    |   29.31   |
> | 5      |              ✔             | Separable $S^2$ | Separable layer norm |    4    |   29.03   |
>
> - Index 1: Naively increasing $L_{max}$ from 2 to 4 and using eSCN convolutions degrade the performance. This is due to overfitting since the QM9 dataset is overall smaller, and each structure in QM9 has fewer atoms, less diverse atom types and much less angular variations than OC20 and OC22.
> - Index 2: Attention re-normalization improves the MAE result.
> - Index 3: Although using $S^2$ activation is stable here (unlike OC20), it results in higher error than using gate activation (Index 2) and the Equiformer baseline (Index 0).
> - Index 4: The proposed separable $S^2$ activation achieves lower error than gate activation and $S^2$ activation.
> - Index 5: The proposed separable layer normalization further improves the result.
>
> Comparing Index 0 and Index 5, the three proposed architectural improvements are necessary to achieve better results than the baseline when using higher degrees on QM9. Overall, these ablation results follow the same trends as OC20.
>
> We have added this ablation study to Table 13 in the revision.
>
> [1] Liao et al. Equiformer: Equivariant Graph Attention Transformer for 3D Atomistic Graphs. ICLR 2023.
>
>
> ---
>
> > 2. [Minor Weakness 1] Update Equation (2) in Appendix A.1.
>
> Thanks! We have updated this.
>
> ---
>
> > 3. [Minor Weakness 2] Move the description before Equation (4) in Appendix A.3
>
> Thanks! We have updated this.

---

### Author Response · Authors · 2023-11-22
**General Response 1: The novelty of the proposed architecture**

# Novelty of the proposed architecture

We would like to reiterate that our work explores how to efficiently scale  equivariant Transformers to use higher degree representations. This question is important because (a) most prior works on equivariant Transformers are limited to small degrees of representations (due to high computational cost) and (b) naively combining Equiformer [1] and eSCN [2] does not outperform the corresponding baselines (Table 1(a) and Table 13 in the revision).

As for novelty, although the proposed architectural modifications are conceptually simple, they are indispensable for scaling Equiformer to higher degree representations. For example, simply using $S^2$ activation as in eSCN leads to training instability (Index 3 in Table 1(a)) and the proposed modification of separable $S^2$ activation is necessary to leverage the more expressive activation function.

Finally, in the same vein as our work, there have been prior works on improving the Transformer architecture in natural language processing (NLP) such as PaLM [3] and LLaMA [4] and in computer vision (CV) like ViT-22B [5]. Following the same backbone Transformer architecture, they propose to use stronger activation functions for better scalability [3, 4, 5], simpler and faster normalization layers [3, 4, 5], and additional normalization for stabilizing training [5]. Each of these modifications is also conceptually straightforward but empirically critical and altogether quite significant for improving performance in NLP and CV. We believe our work proposes similar important architectural modifications for equivariant Transformers and atomistic modeling, backed by thorough evaluations across datasets.

Reference:
[1] Liao et al. Equiformer: Equivariant Graph Attention Transformer for 3D Atomistic Graphs. ICLR 2023.
[2] Passaro et al. Reducing SO(3) Convolutions to SO(2) for Efficient Equivariant GNNs. ICML 2023.
[3] Chowdhery et al. PaLM: Scaling Language Modeling with Pathways. JMLR 2022.
[4] Touvron et al. LLaMA: Open and Efficient Foundation Language Models. ArXiv 2023.
[5] Dehghani et al. Scaling Vision Transformers to 22 Billion Parameters. ICML 2023.

---

### Meta-Review · Area_Chair_Y4xM · 2024-01-02

**Metareview:**

The paper attempts to improve modeling of equivariant symmetries in transformer with application towards atomic system. In this regards, authors build upon Equiformer and scale it higer-degree representations. To achieve this, original tensor product with spherical harmonics is changed to eSCN convolution which can reduce the complexity. Furthermore, to stabilize training attention re-normalization and separable S2 activation were introduced. The paper presents a comprehensive evaluation of the proposed architecture on multiple benchmarks and achieving state-of-the-art results on some. The reviewers were mostly positive about the submission but concerns were raised about the incremental nature of improvements over existing methods. Some reviewers also questioned the generalizability of the techniques and the novelty of the proposed architectural modifications. Author response helped mitigate these concerns.

**Justification For Why Not Higher Score:**

Incremental Advancements: Some reviewers view the improvements as incremental over existing techniques.
Limited Novelty: The architectural modifications, though effective, are seen as derivatives of existing methods.

**Justification For Why Not Lower Score:**

The paper demonstrates significant performance gains across multiple benchmark datasets. Also it introduces many tricks for scaling and stabilizing training which can be useful for the community.

---

### Decision · Program_Chairs · 2024-01-16

Accept (poster)